METHODS

# Visualization and quantification of coral reef soundscapes using *CoralSoundExplorer* software

Lana Minier[1], Jérémy Rouch[2]*, Bamdad Sabbagh[2], Frédéric Bertucci[3], Eric Parmentier[4], David Lecchini[1‡], Frédéric Sèbe[2,5‡], Nicolas Mathevon[2,6,7,8‡], Rémi Emonet[6,9‡]

**1** PSL Université Paris, EPHE-UPVD-CNRS, UAR3278 CRIOBE, Moorea, French Polynesia, **2** ENES Bioacoustics Research Laboratory, University of Saint-Etienne, CRNL, CNRS, Inserm, Saint-Etienne, France, **3** UMR MARBEC, University of Montpellier, CNRS, IFREMER, IRD, Sète, France, **4** Laboratory of Functional and Evolutionary Morphology, FOCUS, University of Liège, Liège, Belgium, **5** Office Français de la Biodiversité, Service Anthropisation and Fonctionnement des Écosystèmes Terrestres, Direction de la Recherche et de l'Appui Scientifique, Gières, France, **6** Institut Universitaire de France, Paris, France, **7** Ecole Pratique des Hautes Etudes, Chart Lab, PSL University, Paris, France, **8** Department of Psychology, University of California, Berkeley, California, United States of America, **9** Université Jean Monnet Saint-Etienne, CNRS, Institut d'Optique Graduate School, Inria, Laboratoire Hubert Curien UMR 5516, Saint-Etienne, France

} Co-first authors.
‡ Co-last authors.
* jeremy.rouch@univ-st-etienne.fr

## Abstract

Despite hosting some of the highest concentrations of biodiversity and providing invaluable goods and services in the oceans, coral reefs are under threat from global change and other local human impacts. Changes in living ecosystems often induce changes in their acoustic characteristics, but despite recent efforts in passive acoustic monitoring of coral reefs, rapid measurement and identification of changes in their soundscapes remains a challenge. Here we present the new open-source software *CoralSoundExplorer*, which is designed to study and monitor coral reef soundscapes. *CoralSoundExplorer* uses machine learning approaches and is designed to eliminate the need to extract conventional acoustic indices. To demonstrate *CoralSoundExplorer*'s functionalities, we use and analyze a set of recordings from three coral reef sites, each with different purposes (undisturbed site, tourist site and boat site), located on the island of Bora-Bora in French Polynesia. We explain the *CoralSoundExplorer* analysis workflow, from raw sounds to ecological results, detailing and justifying each processing step. We detail the software settings, the graphical representations used for visual exploration of soundscapes and their temporal dynamics, along with the analysis methods and metrics proposed. We demonstrate that *CoralSoundExplorer* is a powerful tool for identifying disturbances affecting coral reef soundscapes, combining visualizations of the spatio-temporal distribution of sound recordings with new quantification methods to characterize soundscapes at different temporal scales.

**Data availability statement:** The sound files of the Bora-Bora recording campaign, the configuration files (input file of the processing part of *CoralSoundExplorer*) and the storage files of the generated data (output file of the processing part of *CoralSoundExplorer*) are archived on zenodo.org with DOI 10.5281/zenodo.14577064. The Python scripts used to produce the parametric analysis results presented in the manuscript are included in the same repository on Zenodo.org, as are the Python scripts used to produce the results presented in S2 Text. Version 1.3.0 of *CoralSoundExplorer*, which was used for the work presented here, is available on the GitHub repository: https://github.com/sound-scape-explorer/coral-sound-explorer/archive/refs/tags/v1.3.0.zip.

**Funding:** This research was funded by the University of Saint-Etienne, CNRS, Inserm, Saint-Etienne Métropole, Ecole Pratique des Hautes Etudes, Institut Universitaire de France (NM and RE), and Labex CeLyA. The funders had no role in study design, data collection and analysis, decision to publish, or preparation of the manuscript.

**Competing interests:** The authors have declared that no competing interests exist.

## Author summary

Passive acoustic monitoring is an increasingly popular method for monitoring ecosystems, but analyzing the data generated by this approach remains complex. In this paper, we present *CoralSoundExplorer*, a tool for analyzing large sets of sound recordings. The tool is open source and has been designed to be easy to use, even by non-specialists, thanks to a graphical interface. This graphical interface is also available in an online version to allow users to visualize previously processed data without needing to install the software. The *CoralSoundExplorer* software transforms recordings of coral reef soundscapes into visual representations using UMAP embeddings coupled with tools for quantifying phenomena. By projecting sound samples into acoustic spaces, *CoralSoundExplorer* enables the observer to grasp the characteristics of the soundscapes, their differences and similarities and their organization on different temporal scales. These acoustic spaces and their temporal dynamics can be quantified, for example to account for the speed at which soundscapes change over time or to identify clusters of sound samples on the basis of their acoustic similarities. We use an example of the reef soundscapes of the island of Bora-Bora to present and illustrate the functionalities of our software, confirm previous findings, provide new insights, and demonstrate the applicability of the software for the analysis of large datasets. Based on the Bora-Bora dataset, we also provide a parametric study of *CoralSoundExplorer* to show the effects of its different settings on the analysis results. Detailed instructions for installing and using the software are also provided in the supplementary materials S3 Text.

## 1. Introduction

In the current context of increasing natural habitat degradation, assessing ecosystem biodiversity and monitoring its spatio-temporal dynamics at various scales is an absolute necessity for implementing effective conservation policies, particularly for biodiversity hotspots such as tropical forests and coral reefs [1–6]. This requires rapid and reliable methods for processing large amounts of data [7]. Over the past decade, Passive Acoustic Monitoring has emerged as a crucial tool for monitoring terrestrial and marine environments, as soundscapes, at least for their biophony part, are reliable proxies for biodiversity [8–11]. By utilizing sounds as cues to the presence and activity of organisms, Passive Acoustic Monitoring provides a non-invasive, cost-effective approach to long-term monitoring of biodiversity while allowing high temporal resolution [12–15].

Coral reefs host some of the highest levels of biodiversity on the planet and provide many ecosystem services, direct or indirect benefits to several hundred million people worldwide: coastal protection, building materials, food and income from fishing or tourism [16]. However, most coral reefs are currently experiencing severe

declines under the impacts of global change [17–19]. To obtain the information needed to deploy effective conservation and management policies, it is essential to have tools for spatio-temporal monitoring of the biodiversity of these reefs [20,21]. Since coral reefs are highly sonorous worlds due to the activity of numerous organisms [15,22–25], Passive Acoustic Monitoring is emerging as a particularly interesting tool for monitoring biodiversity and its dynamics [26,27]. The sounds produced by many fish species can provide information on their reproductive activity and enable changes in population abundance to be identified [28]. Sounds produced by invertebrates can also serve as indicators. For example, the density of snaps produced by shrimps is correlated to the oxygen content of the water [29]. It has also been shown that close but distinct seascapes (e.g., fringing reef, barrier reef, outer slope) are characterized by different soundscapes [26,30,31]. The soundscapes of coral reefs therefore provide information about the conditions of the habitats and communities they harbor [14,26,27,32]. Moreover, they do not provide a static image but can reflect changes over time, at different temporal scales [15,33]. As a sort of multidimensional spatial and temporal picture, coral reef soundscapes are thus witnesses to biodiversity and species richness [28,34–36].

While the benefits of using Passive Acoustic Monitoring for coral reefs are commonly admitted with well-developed recording techniques at cost fairly acceptable, there remains a major challenge: developing effective analysis tools capable of apprehending large amount of data in a reasonable time and providing both qualitative and quantitative information easy to interpret [17,37]. Manual exploration of soundscape recordings by visual exploration of spectrograms and recording tracks listening is extremely time-consuming and observer-dependent [8,38,39]. The gain brought by the Passive Acoustic Monitoring approach in terms of ease of data accumulation (hydrophones are easy to install and quite inexpensive) is thus frequently diminished by an analysis process just as time-consuming and tedious as traditional survey methods (video footage, photography, and visual census).

Soundscapes are commonly described using acoustic indices [40]. In terrestrial environments, a chosen set of acoustical indices can be used to perform statistical data analysis in order to infer ecological results. Correlation studies and dimension reduction techniques can be applied to sets of acoustic indices to explore differences between soundscapes and their evolution. From a set of indices, clustering methods can be used to perform semi and unsupervised analysis using machine learning tools. Such approaches have already been applied to coral reef soundscapes, using for example the Acoustic Complexity Index (ACI) [41], Acoustic Entropy (H) [42], Acoustic Diversity Index (ADI) [43]. The ACI measures the average temporal variations in frequency band levels [44] and is assumed to correlate with the ratio of biotic to anthropogenic sounds. ADI measures the Shannon entropy of level values over several frequency bands, supposed to be related to the diversity of sound sources over the corresponding frequency band(s) [45]. Acoustic entropy measures the Shannon entropy of the amplitude envelope and/or spectra of sound signals [12] and is related to the temporal and spectral diversity of sounds. However, a growing body of experimental evidence suggests that terrestrial acoustic indices are not ideal for characterizing underwater soundscapes, and may even perform poorly for specific terrestrial soundscapes [5,30,46], as differences in sound propagation phenomena, source types, spectral characteristics and temporal dynamics lead to ecosystem-specific organization and composition of soundscapes. Recent approaches use deep learning techniques to get a set of sound descriptors from artificial neural networks [47]. Unlike acoustic indices, the descriptors given by a neural network are not easily intelligible, but because of their efficiency in many sound analyzing tasks and their capacity to process fast on big database, deep learning approaches applied to soundscape analysis has led over the past few years to several processing workflow papers publications for a large set of bioacoustics and ecoacoustics problematics [7,9,48–54]. Among the scientific publications using artificial intelligence in the framework of ecoacoustics, the paper of Sethi et al. (2020) [55] inspired the present work. These authors proposed to analyze natural soundscapes using the pretrained VGGish neural network [56]. From the 128 features per sound sample given by this neural network, a non-linear dimension reduction down to two dimensions was performed, allowing the visualization of the sound samples constituting the soundscapes in an acoustic space where similar sounds are mostly close to each other while different sounds are mostly distant to each other. This approach enabled visual exploration of soundscape phenology. While the analysis described in Sethi et al (2020) [55] was appealing, it remained limited to visual inspections of soundscapes. One

of our aims is to complement this approach with a quantitative method that provides reliable measures of the proximities and differences between the elements composing the soundscapes. Our other aim is to provide the user with an intuitive and user-friendly tool, where Sethi et al. (2020) [55] only offered a workflow indicating how to use a set of existing or specially developed programming language packages. We also wanted to provide a flexible tool where different settings and processes could be easily handled and tested. For instance, Sethi et al (2020) [55] used the VGGish embedding and not simpler non-deep learning embeddings (e.g., the mel-spectrogram).

Here we introduce a new, powerful analytical tool with a graphical interface designed with the objective of being intuitive and user-friendly, named *CoralSoundExplorer* [57]. It uses machine learning techniques to analyze, describe, and quantify underwater soundscapes. The *CoralSoundExplorer* workflow uses an initial acoustic embedding that can be either obtained from a Convolutional Neural Network (CNN), a mel-spectrogram or a mel-spectrum, followed by a Uniform Manifold Approximation and Projection (UMAP, [58]) to replicate the geometric relationships of the original high-dimensional space in lower dimensions. Using these machine learning approaches, *CoralSoundExplorer* automatically extracts relevant acoustic features from underwater sound recordings, providing valuable information for biodiversity assessment, habitat monitoring and understanding the temporal dynamics of coral reef ecosystems to identify disturbances.

As illustrated in Fig 1, this paper is organized in a succession of sections as follows. In section I, we present the general characteristics of a coral reef soundscape and describe the recordings made on the island of Bora-Bora, which are used as sample data in this article. In Section II, we outline the principle of sound signal processing by *CoralSoundExplorer* software, the graphical representations and analyzes it enables, and the metrics that can be measured. In Section III, we show, as an

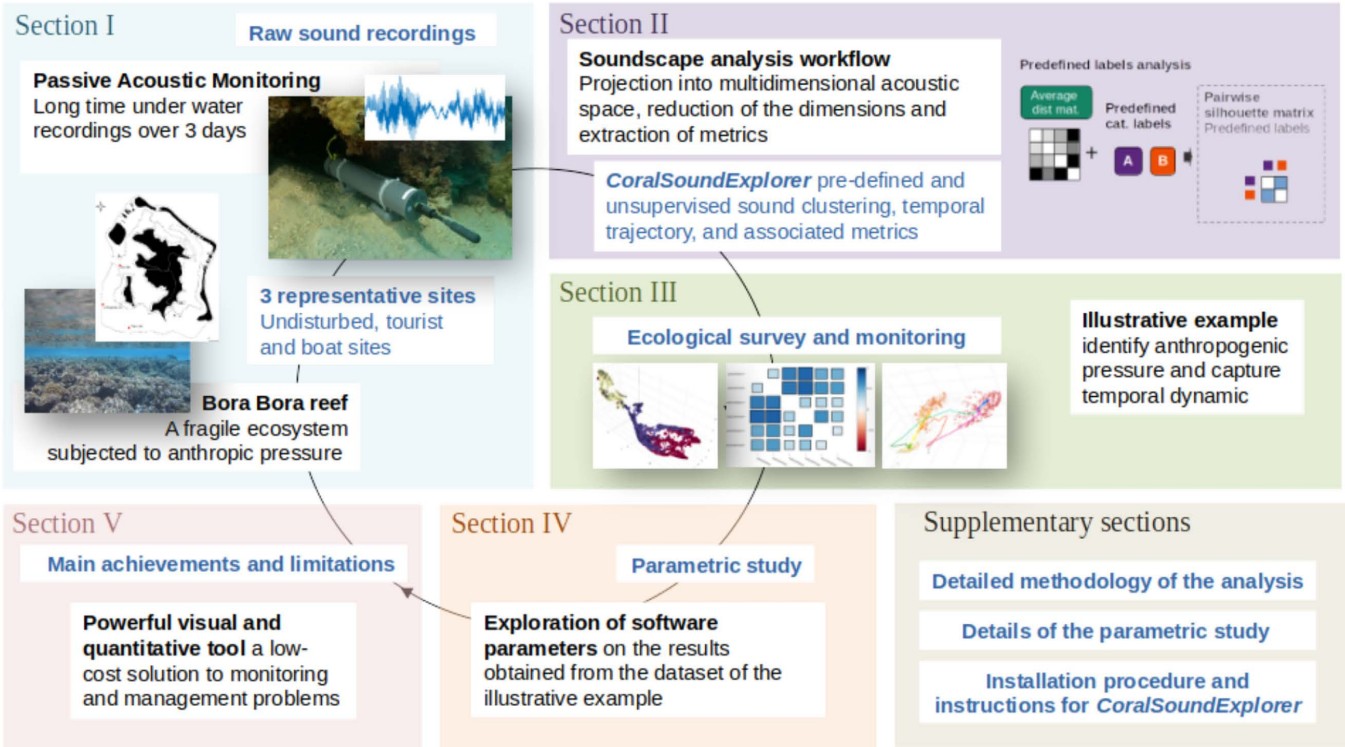

**Fig 1. Study framework.** Section I: General characteristics of a coral reef soundscape and presentation of the recordings made on the island of Bora-Bora that serve as example data in the present study. The base map was reproduced from [59]. Section II: Presentation of *CoralSoundExplorer* software (analysis workflow, graphical output and measurable metrics). Section III: Results obtained with *CoralSoundExplorer* from the sample recordings made on Bora-Bora. Section IV: Parametric study of *CoralSoundExplorer* settings using the Bora-Bora dataset. Supplementary sections: S1 Text: Detailed Methodology of *CoralSoundExplorer*, S2 Text: Parametric study of *CoralSoundExplorer*, S3 Text: Software installation procedure and instructions.

illustrative example, the results obtained by *CoralSoundExplorer* with sound recordings made on the coral reef of the island of Bora-Bora. We explain how *CoralSoundExplorer* can be used to quickly and easily identify soundscapes according to the anthropogenic pressure or other perturbations they are subject to, and to capture their temporal dynamics. We describe how the analyses carried out with *CoralSoundExplorer* make it possible to account for these differences and dynamics. In section IV, we present a parametric study of the software. Using the Bora-Bora dataset, we explore the effects of the initial acoustic embedding and those of the dimension reduction process and its parameters. Section V examines the main achievements and limitations of *CoralSoundExplorer*, and presents our recommendations for exploring reef recordings with our software. We have added three supplementary sections to the document, S1, S2 and S3 Texts. The first complements Section II by providing detailed information on the software workflow and the proposed analysis, thus offering a more in-depth explanation of the methodology. The second completes section IV by providing figures and raw descriptions of the parametric study results. The third supplementary section, S3 Text, describes the software structure and installation procedure.

## 2. The soundscapes of coral reefs and the example of Bora-Bora Island

### 2.1. What is a coral reef soundscape?

Like all aerial or underwater soundscapes, those of coral reefs are the result of an interweaving of sounds produced by animals (biophony - Fig 2A and 2B), sounds from human activities such as shipping (anthropophony - Fig 2C), and sounds emanating from geophysical processes such as waves breaking on the reefs (geophony - Fig 2D). As coral reefs are among the world's richest ecosystems in terms of biodiversity [60,61], their soundscapes are among the most complex of aquatic realms. Fish make sounds when they eat or swim and most of them produce sounds to communicate, notably during agonistic interactions, in response to threats such as the presence of predators, or during courtship and spawning [23] (Fig 2B). While up to half of the fish families may have sonorous species, benthic invertebrates also play a major role in coral reef biophony [62–64]. Snapping shrimps, common on coral reefs, are very noisy [62]. Bivalves, clawed and spiny lobsters and burrow-dwelling mantis shrimps also produce sounds [65]. Sea urchins when moving or grazing also contribute to biophony [24,66]. As coral reefs vary according to depth, hydrodynamic conditions and other parameters, there is a wide diversity of animal species assemblages and associated habitats [30]. This ecological diversity drives a diversity of soundscapes that reflect the properties of the coral reef ecosystem, in its spatial and temporal components, and are thus witnesses to the diversity of animals, as well as coral cover [14,26,27].

### 2.2. The example of Bora-Bora: Reefs under heterogeneous anthropic pressure

Bora-Bora, a tropical volcanic island in French Polynesia (16°29' S, 151°44' W), is an excellent example of several coral reefs habitats: fringing reef, channel, barrier reef, mangrove, *etc.* [67]. Bora-Bora is surrounded by a coral reef of 70 km² which is home to a high diversity of fish, invertebrates and other marine organisms [3]. Despite being a renowned international tourist destination with luxurious resorts and environmental commitment, it faces anthropogenic pressures, mainly from intense tourism [3,68,69]. This tourism success comes with a price, as it generates significant maritime traffic, inevitably accompanied by noise pollution caused by the intensive use of internal combustion engine-powered boats. To regulate the presence of tourist activities in the lagoon, 14 ecotourism sites have recently (2019) been identified by the town council and the tourism committee [3,70]. Despite these measures, some sites on Bora-Bora remain heavily impacted by tourist activities, not to mention the fishing activities that meet the food needs of a large part of the population [3,71]. To give an idea of the scale of nautical activity, 711 boats for personal use such as sailing, leisure and fishing are based on Bora-Bora [72].

### 2.3. The Bora-Bora coral reef dataset

We selected three sites in the south-western part of Bora-Bora lagoon (Fig 3): a tourist site (16°32'11.543" S, 151°43'30.575" W) used for snorkeling activities with healthy corals and a high fish species richness (max. 2 m depth),

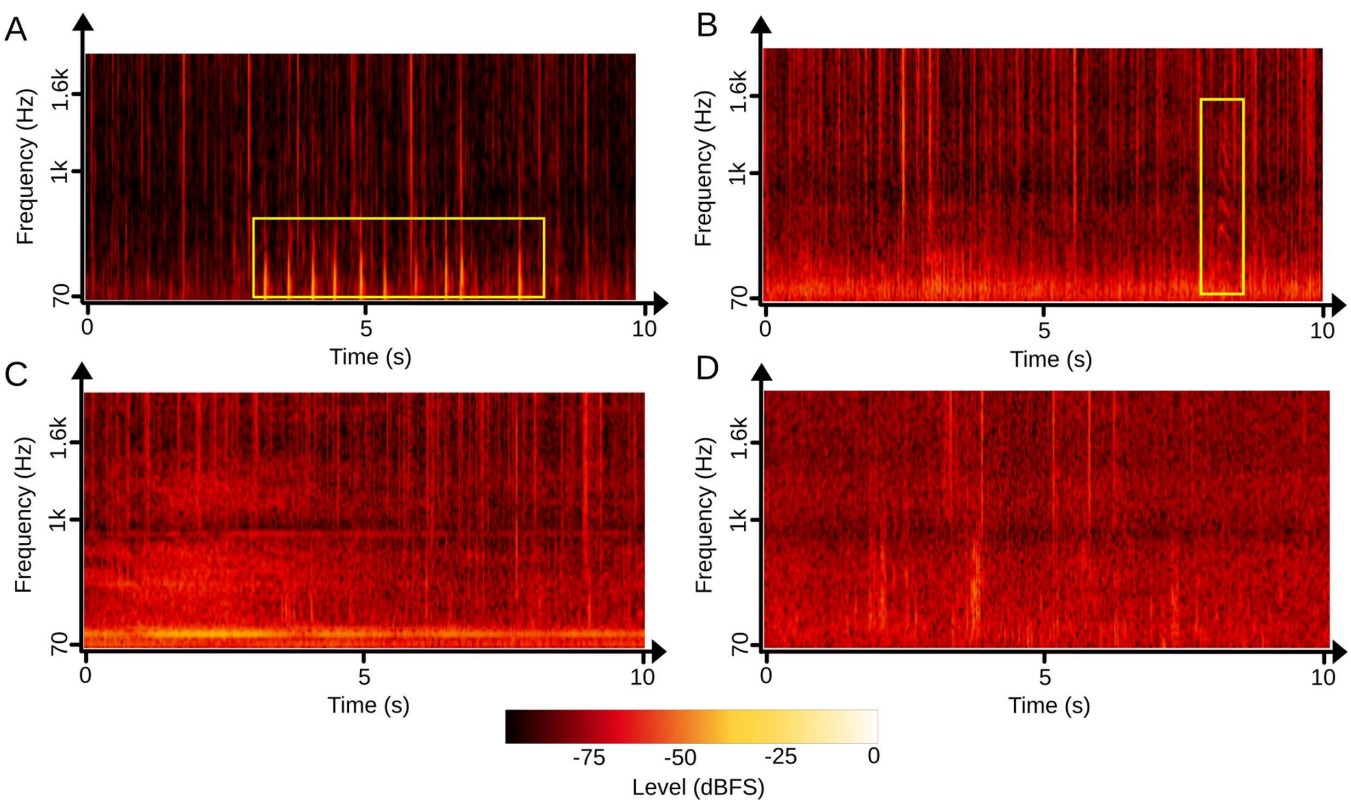

**Fig 2. Soundscapes diversity in a coral reef. (A)** Spectrogram of a reef soundscape recorded during the day. Fish sounds (yellow square) are mostly concentrated at low frequencies (< 1 kHz). **(B)** Spectrogram of a reef soundscape recorded at night showing an overall increase of sounds levels mostly due to an increase in the number of snapping shrimp sounds. An unidentified fish sound is also present (yellow square). **(C)** Spectrogram showing the broadband noise of a passing boat, with higher intensity during the first 5 seconds. **(D)** Spectrogram showing broadband masking effect caused by rain. Spectrograms were plotted with *CoralSoundExplorer*, using FFT window size = 2048 samples, sampling frequency = 44100 Hz.

an undisturbed site (16°31'46.956" S, 151°47'19.823" W) where very few boats can access due to a high coral cover and which has no tourist activity, and a boat site where boat traffic is particularly intense in a deep sandy channel (16°30'7.416" S, 151°46'5.448" W) [3,15]. We carried out three passive acoustic recording sessions in November and December 2022. The duration of the recordings was deliberately limited to enable parallel manual analysis of the sound recordings and comparison with the results obtained with *CoralSoundExplorer* software. This dataset was therefore suitable for checking the results obtained with *CoralSoundExplorer*.

On each of the three sites and for three different days (November 16, November 22 and December 12), we deployed simultaneously SNAP long-term autonomous acoustic recorders (Loggerhead Instruments; Sarasota, FL, USA) equipped with a HTI-96-Min hydrophone (sensitivity 169.9 dB, 170.0 dB, and 170.1 dB re 1 V at a sound pressure of 1 µPa; flat frequency response from 2 Hz to 30 kHz). The recorder positioned at each of the three sites recorded the soundscape over 24 h, from 12:05 p.m. to 12:05 p.m. (included) the next day, with a recording cycle of 1 min every 10 min (sampling frequency 44.1 kHz; 16-bit resolution). On all three sampling dates, the three recorders were systematically positioned in exactly the same spot, on a sandy substrate beneath a coral head, at depths of 1 m, 2 m and 5 m for the tourist, undisturbed and boat sites respectively.

The original recordings were high-pass filtered with an 8-order zero-phase Butterworth filter set at a cut-off frequency of 70 Hz. The soundscape analyses were carried out in the low-frequency band of interest, between 70 Hz and 2 kHz

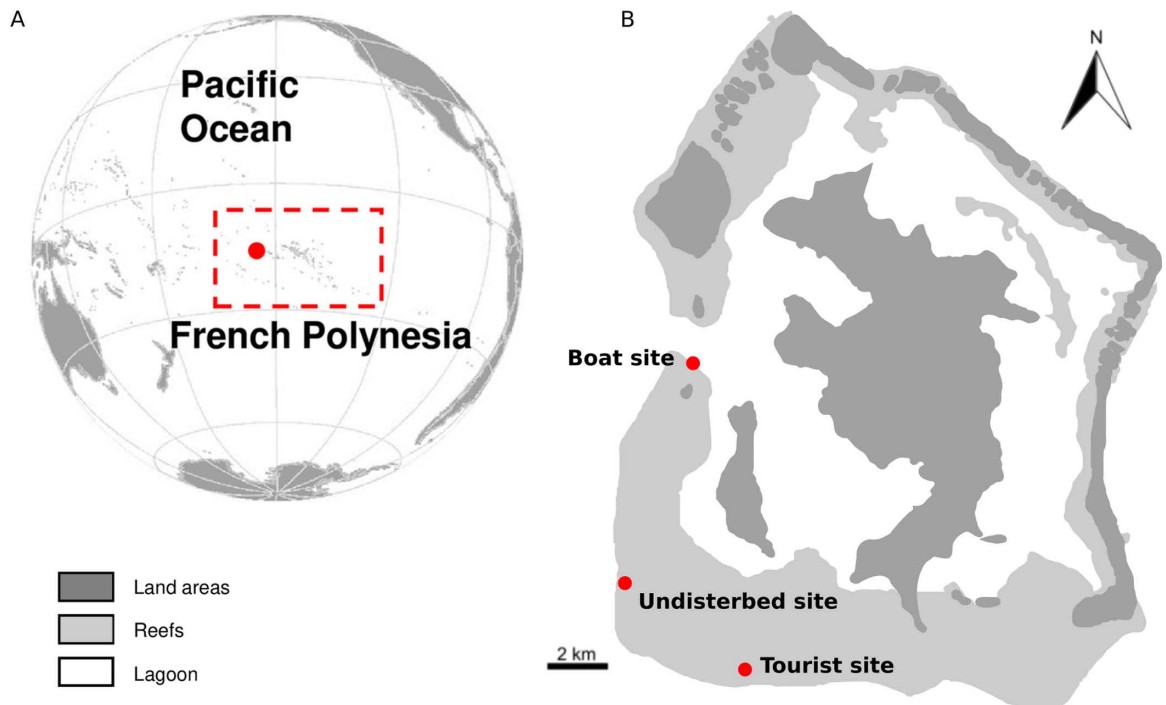

**Fig 3. Locations of French Polynesia and the recording sites on the Bora Baro Island. The base map was reproduced from [59]. (A)** World map showing the location of the French Polynesia archipelago (red square) and the world distribution of coral reefs area (blue square). **(B)** Map of Bora-Bora showing the three recording sites (boat site, undisturbed site, and tourist site). In dark grey: land areas; in light grey: reef area.

where most fish sounds and anthropogenic noises are found [73,74]. All recordings were labeled according to location (undisturbed, tourist or boat sites), time of day (daytime or nighttime, depending on the exact time of sunrise and sunset), and date (replicate 1 for November 16, replicate 2 for November 22, and replicate 3 for December 12). A composite label composed of all three previous labels has been created to enable simultaneous and combined analysis of these three factors. This combination of initial labels produced 18 composite labels (3 x 2 x 3). We also listened to all these recordings, associating them with metadata such as the presence of rain or motorboat noise.

## 3. *CoralSoundExplorer* software: Analysis workflow, graphical output and measurable metrics

The soundscapes analysis workflow handled by *CoralSoundExplorer* [56] consists of three main steps: the projection of sound samples into a multidimensional acoustic space, the reduction of the dimensions of this space so that it can be visualized in 2D or 3D, and the extraction of metrics characterizing the organization of the sound samples composing the soundscapes. This part II briefly presents the *CoralSoundExplorer* analysis workflow and the results that can be drawn from it. More technical details are given in supporting information S1 Text.

### 3.1. Soundscape analysis workflow

**3.1.1. From original sound signals to graphical visualization in acoustic space.** *CoralSoundExplorer*'s workflow is illustrated in Fig 4, from original sound signals to graphical representations and possible calculations. The first part of the workflow (Fig 4A) is partly inspired by previously published work [55]. The original one-minute raw recordings are first divided into one-second signals. These one-second signals are projected into an initial multidimensional acoustic space

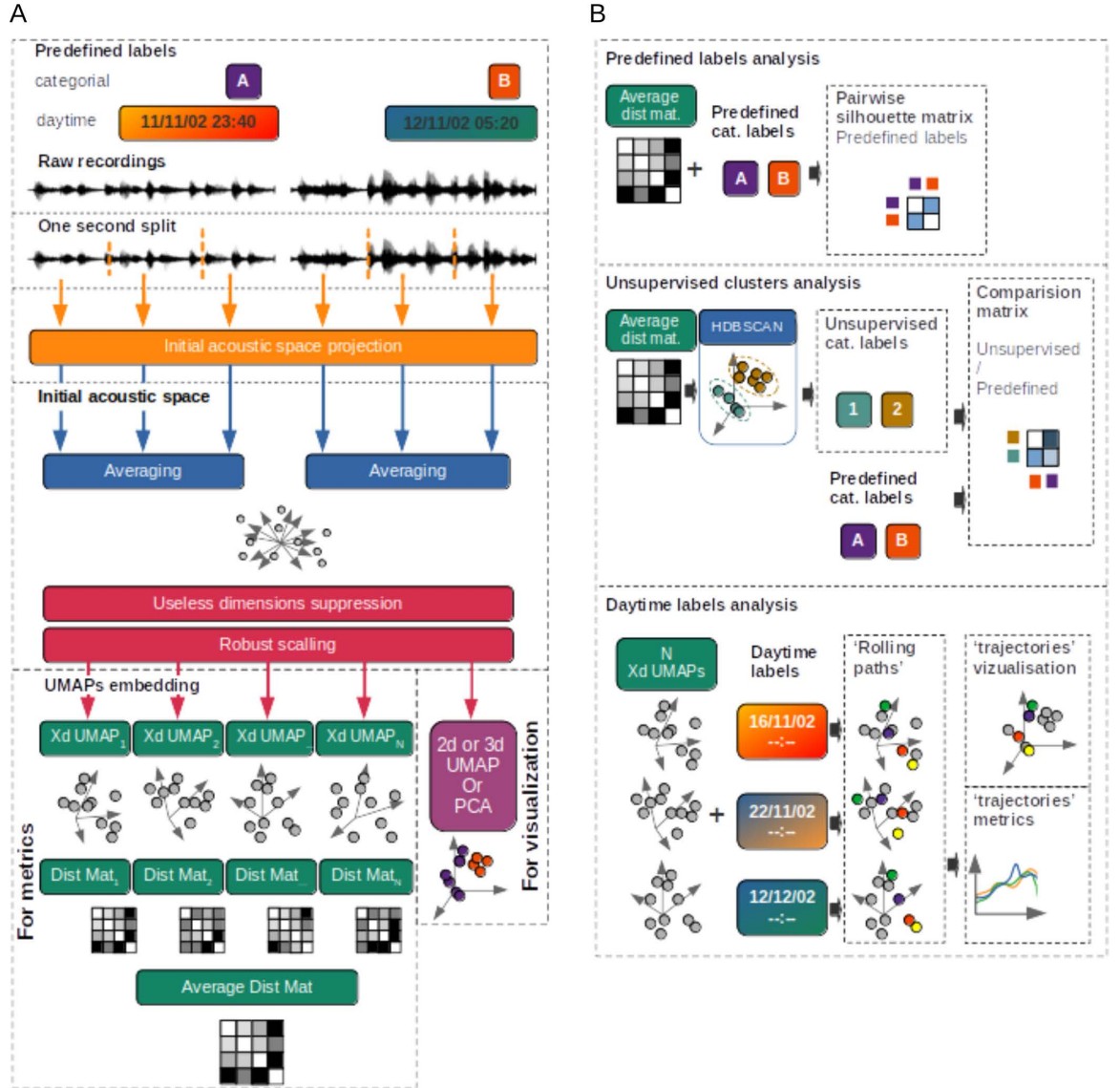

**Fig 4. *CoralSoundExplorer* workflow. (A)** From sound recordings to graphical representations and distance matrices. Raw sound recordings, associated with predefined discrete (categorical) or continuous labels (e.g., recording locations and times), are cut into one-second extracts and projected into an acoustic space (using mel-spectrum, mel-spectrogram or VGGish embedding). These one-per-second samples are then aggregated according to the chosen integration time (15 seconds for Bora-Bora's study case) and their acoustic descriptors are averaged over this duration. Dimensions containing no information (i.e., descriptors equal to 0 or constant for the whole dataset) are eliminated. This step is followed by a normalization process on each remaining dimension (robust scaling). The population of aggregated sample descriptors is projected into a two- or three-dimensional space using the UMAP or PCA method for visualization purposes. A UMAP (Xd) operation with a fixed dimensionality (X) is parallelly repeated N times. For each iteration, the distances between each pair of sounds in the acoustic space are calculated to establish distance matrices (Dist Mat$_n$ with $1 \leq n \leq$ **N)**, which are then averaged (Average Dist Mat). **(B)** Analyses based on UMAP embeddings. Top panel: Evaluation of the suitability of predefined labels to describe the organization of the acoustic data. This evaluation can be made visually from the 2/3D UMAP or PCA of the initial acoustic space, where each sound is identified by a colored dot according to a predefined label. It is also quantified from the average distance matrix by calculating silhouette indices (Pairwise Silhouette index). Middle panel: Unsupervised sound clustering. The HDBSCAN algorithm identifies sound clusters based on their proximity in the average distance matrix. These clusters are associated with so-called unsupervised labels (Unsupervised cat. labels) that identify sounds in acoustic space. The unsupervised clusters can be visualized in a 2/3D projection (visual UMAP or PCA) and compared with clusters derived from predefined labels (Predefined cat. labels) to assess the correspondence between the two categories of labels (Unsupervised/Predefined comparison matrix). Bottom panel: The temporal trajectories of soundscapes in sound space (Rolling path) are calculated and plotted based on the position of UMAP points relative to the starting position (recording start time). This calculation is averaged over the N UMAPs (see text for details).

or embedding. This projection can be performed using a mel-spectrum transform, a mel-spectrogram transform, or the embedding from the VGGish neural network.

If the mel-spectrum transform is used, each dimension in the multidimensional acoustic space represents the energy values of one of the 64 mel-frequency bands covering the entire frequency range of interest. This spectrum is computed over a one-second interval, and the energy values for each frequency band are converted to their natural logarithms. If the mel-spectrogram transform is selected, the one-second signals are processed using a Short-Time Fourier Transform (STFT), resulting in a two-dimensional time-frequency representation of the sound. The same mel scale as for the mel-spectrum is used but the short time spectrums are computed over 2048 samples with a hop size of 0.01 seconds. Working with a sampling frequency of 44100 Hz leads to a hop size of 441 samples. The resulting mel-spectrogram contains 6400 time-frequency energy values, which are also transformed into their natural logarithms. Each of these values is treated as a single dimension in the acoustic space. For the VGGish embedding, the first 0.96 s of the mel-spectrograms described above (one per second of raw audio thus the remaining 0.04 s of those spectrograms are not used. See details in S1 Text) are fed into a Convolutional Neural Network (CNN) pre-trained on a large YouTube's audio-video dataset named VGGish [75]. The acoustic space is the output of this CNN where each second of recording is assigned 128 dimensions related to its VGGish acoustical characteristics. Once the entire dataset has been projected into the mel-spectrum space, the mel-spectrogram space or the VGGish embedding, a process of deleting unnecessary dimensions (i.e., dimension with constant values over the whole dataset, thus dimensions containing no discriminative information) is carried out followed by a normalization process (robust scaling) on each remaining dimension. A temporal grouping of the individual projections for each second can be performed by calculating the average position in the initial acoustic space of a positive integer number of consecutive seconds in each recording, without overlap. This temporal grouping parameter is expressed in seconds and called "integration time".

A dimension reduction process is carried out from the initial acoustic space using either an implementation of the UMAP algorithm or a PCA, to obtain a sufficiently low number of dimensions (2 or 3 dimensions) to allow visualization of the acoustic space.

The UMAP transformation process is not deterministic and no individual UMAP is *a priori* better than another. This stochasticity involves differences in point positions between two different UMAPs projections. Metrics calculated directly from these projections or from point distance matrices are also affected by this stochasticity. Thus, to mitigate the random effects of the UMAP algorithm and reduce the uncertainties of the analysis metrics, the UMAP dimension reduction process is repeated. From these repeated UMAPs and the average distance matrix calculated from them, different sound organization metrics are finally calculated.

### 3.2. Analysis

#### 3.2.1. Analysis 1: Assessing the suitability of predefined labels for describing data organization.
The three types of analysis carried out on the basis of UMAP embeddings are illustrated in Fig 4B. The corresponding *CoralSoundExplorer's* display features are illustrated in Figs 5–7.

In the first analysis (Predefined labels analysis; Figs 4B top and 5), we assess the extent to which sounds that all share the same predefined label (e.g., having all been recorded at the same location or all on the same day or same day period) effectively form an identifiable and distinct group from sounds with other labels. The assessment of this ability of predefined labels to form distinct acoustic clusters is based on the calculation of the silhouette index [76]. This index is used here to quantify the overlap of positions occupied in acoustic space by sounds bearing two different labels. The silhouette index gives an indication of the dissimilarity of sounds belonging to two labels. When the silhouette index equals 1, sounds defined by the two labels from two distinct groups in acoustic space. When the silhouette index is equal to 0, sounds belonging to both labels form undifferentiated groups, meaning that the labels assigned to the sounds do not translate into differences in position in acoustic space. The points representing the sounds of the two labels then occupy

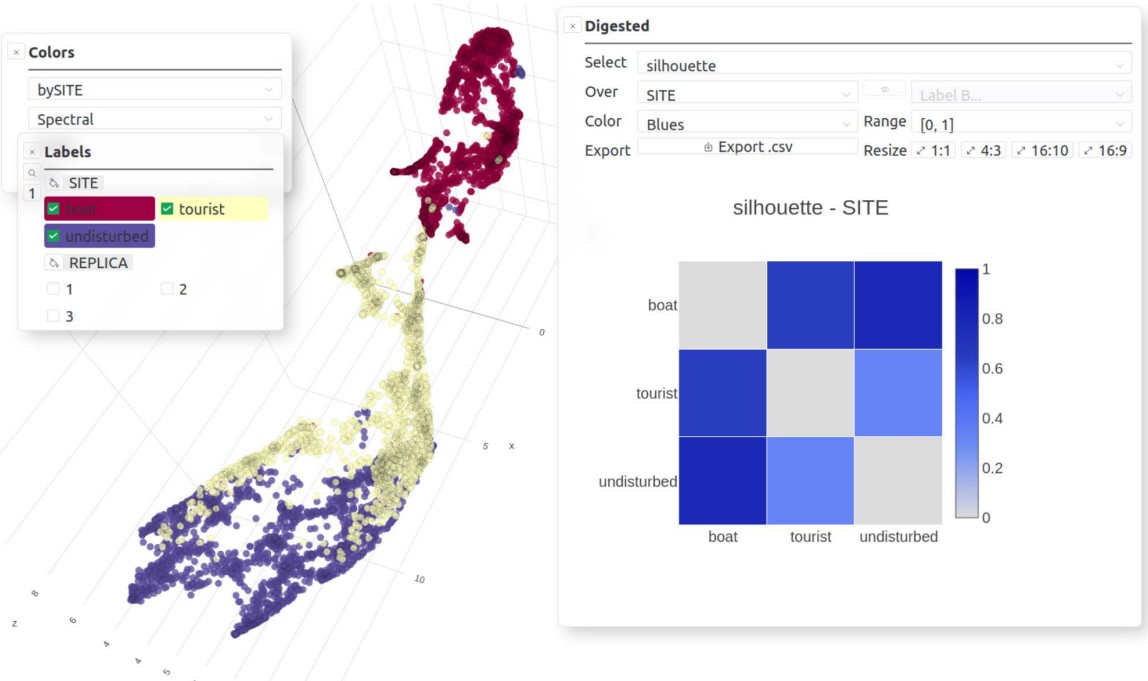

**Fig 5. Analysis offered by *CoralSoundExplorer*: Assessing the suitability of predefined labels for describing data organization (here 3D UMAP view).** Graphical display possibilities: Color points (sounds) according to the values of a categorical variable. Pairwise silhouette indices matrix for one categorical variable. Export possibilities: UMAP/PCA plots (.png,.svg), Pairwise silhouette indices matrix (.csv) and its plot (.png,.svg).

the same region of acoustic space. For a given categorical label, *CoralSoundExplorer* automatically calculates the silhouette indices between the sounds considering each of the label's categorical values in pairs. The results are displayed as a matrix of colored squares, the color indicating the index value. This allows visual exploration of how sounds share acoustic space according to the predefined labels. The numerical values of the matrix are exportable as a table in a CSV file.

### 3.2.2. Analysis 2: Unsupervised sound clustering.

In this second analysis (Figs 4B middle panel and 6), *CoralSoundExplorer* seeks to identify sound clusters in an unsupervised way based solely on N UMAPs average distance matrix, without prior reference to the labels previously assigned to the sounds. This search for clusters is made using the HDBSCAN clustering algorithm [77]. This algorithm identifies groups of sounds (unsupervised clusters) in the acoustic space within which the sounds are similar to and different from those of other groups. This algorithm has the advantage of working without prior knowledge of the number of clusters. It can also exclude some sounds from any cluster if it judges them to be too isolated in the acoustic space. The result can be directly visualized on the UMAP acoustic space representation in two or three dimensions. The points representing the sounds are colored according to their cluster membership. These unsupervised groups, defined automatically by the HDBSCAN algorithm, can be compared with a predefined label, in order to identify the proportion of each of its categories making up each unsupervised group. This sub-analysis is presented in the form of a matrix, with the rows corresponding to the labels of the unsupervised clusters, the columns corresponding to the categorial predefined labels. Each value in this matrix is the rate of the corresponding predefined label category over the entire set of points in the corresponding unsupervised cluster. To allow visual assessment of these values, *CoralSoundExplorer* offers a graphical representation with colored squares, whose shades of color are proportional to the matrix values. HDBSCAN's computation parameters are preset by default in the software. They can be adjusted to obtain different levels of analysis refinement, leading to different sizes and number of unsupervised clusters.

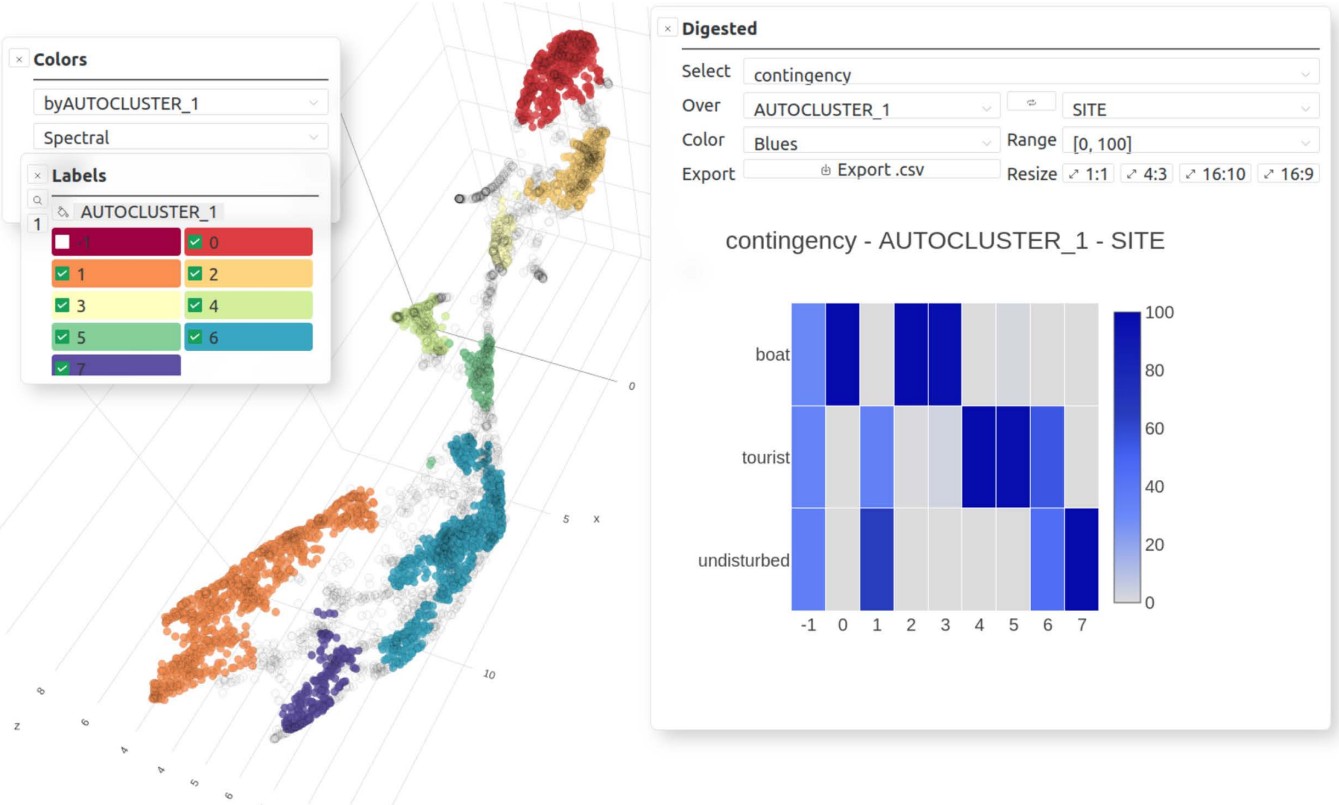

**Fig 6. Analysis offered by *CoralSoundExplorer*: Unsupervised sound clustering (here 3D UMAP view).** Graphical display possibilities: Color points (sounds) according to the values of unsupervised clusters id (cluster label -1 is related to unclassified sound samples), Contingency matrix (i.e., percentage of each category of one categorical variable within each category of another categorical variable). Export possibilities: UMAP/PCA plots (.png,.svg), contingency matrix (.csv) and its plot (.png,.svg).

**3.2.3. Analysis 3: Temporal trajectory of soundscapes.** The third analysis tracks the evolution of the soundscape over time (Figs 4B bottom panel and 7). It provides a mapping visualization and associates a metric to quantify the evolution of sounds over time. Temporal monitoring of the soundscape involves defining the sub-regions of acoustic space explored by sounds over a given period and for a given recording location. This creates 'paths' in the acoustic space along time points. These paths are computed from the UMAP or PCA projection in 2 or 3 dimensions. They are plotted graphically in this 2/3D representation to visualize the phenology of the soundscape. These trajectories are colored according to the time of day. By comparing the temporal paths of the same recording location between different days, these paths make it easy to detect outliers in the temporal trajectories of the soundscape that may indicate changes in the ecological situation (e.g., between days). If the 2D or 3D UMAP plot is used, the calculations enabling this visualization of temporal paths are originally based on a single projection and are therefore subject to UMAP stochasticity. To give a stable metric of the evolution of the soundscape over time relative to each recording site, *CoralSoundExplorer* makes use of the N UMAPs. The measure is the distance from the average starting point of each site's paths. Because this distance in UMAPs is not absolute, it is defined relative to the average distance between its reference point (starting point) and its 100 nearest neighbors in the UMAP. For a given site, day and time, the median value of these distances (over the N UMAPs) is retained together with a measure of the dispersion of these distances, the interdecile range between the first and the ninth decile. These median distances and their dispersions are then plotted against time of day, for a given site

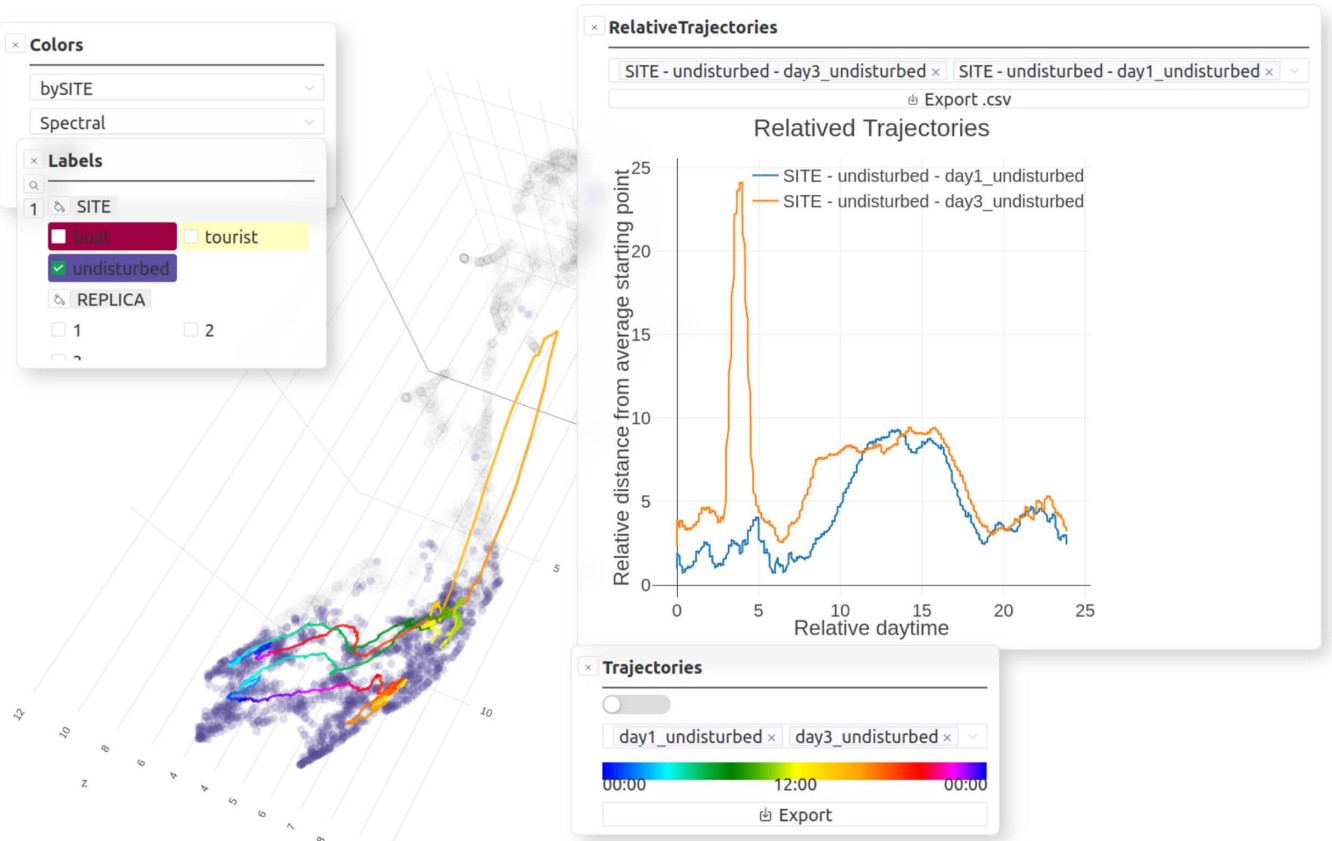

**Fig 7. Analysis offered by *CoralSoundExplorer*: Temporal trajectory of soundscapes (here 3D UMAP view).** Graphical display possibilities: plot predefined time trajectories, plot "Relative distances [from average starting point]". Export possibilities: UMAP/PCA plots (.png,.svg), "Relative distances [from average starting point]" (.csv) and their plots (.png,.svg).

and a given replicate. In this way, it is possible to appreciate phenology in a different way from that possible directly on the UMAP or PCA visualization. All these relative trajectories start and often end with a value around one indicating a cyclic phenology of soundscapes during the whole day/replicate. During the day, the relative trajectories can diverge, indicating a change of soundscape compared to the one present during their starting point.

### 3.3. Other display features and data exploration tools provided by *CoralSoundExplorer*'s user interface

*CoralSoundExplorer* is a tool for exploring soundscapes, combining a graphical interface for visualizing the distribution of recorded sounds in a 2- or 3-dimensional acoustic space, with functions for quantifying this distribution. *CoralSoundExplorer*'s graphical interface includes interesting possibilities for exploring a sound database. *CoralSoundExplorer* offers the option of coloring points (corresponding to the different sounds) directly from the graphical interface, according to the values taken by a predefined categorical variable or by the cluster obtained after unsupervised cluster identification. It is also possible to hide sounds in the projection space according to the value(s) of another or the same categorical variable (Fig 8A). This allows the user to quickly visualize the organization of recorded sounds according to the attribute value of a categorical variable, and to visualize possible interactions between two or more categorical variables.

Each sound sample corresponding to a point in the UMAP/PCA 2D or 3D acoustic space can be played by clicking on it (Fig 8B). The spectrogram of this signal is also displayed. The size of the spectrogram analysis window can be

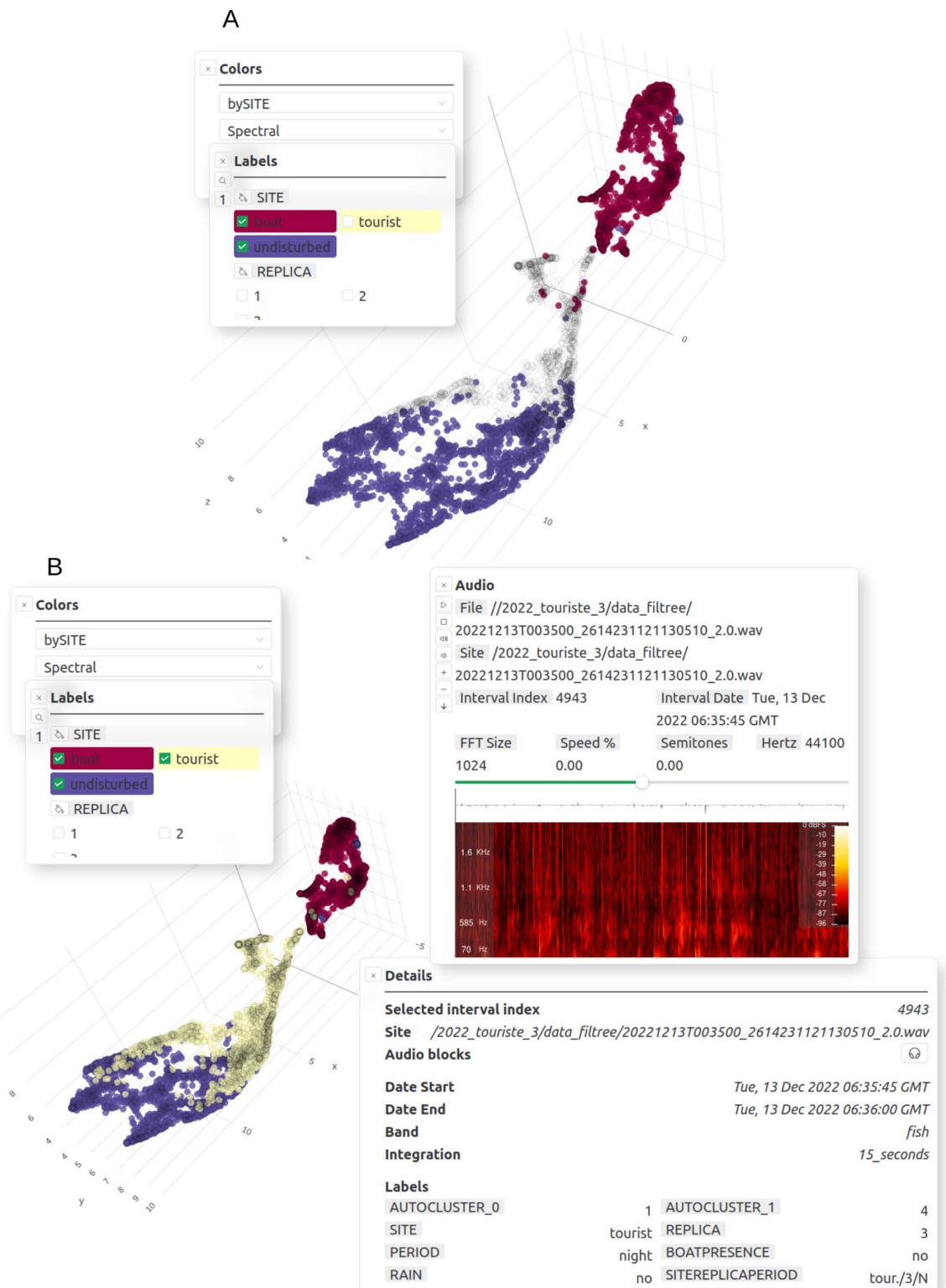

**Fig 8. Other display facilities and data exploration tools provided by *CoralSoundExplorer* 1 (3D UMAP view). (A)** Display features: color points (sounds) according to the values of a categorical variable, highlight points according to the values of a categorical variable (the same or another). Export possibilities: UMAP/PCA plot (.png,.svg). **(B)** Display features: sounds' information and their spectrograms after clicking on the corresponding point. An audio player is provided to listen to the selected sound. Export possibilities: sound (.wav).

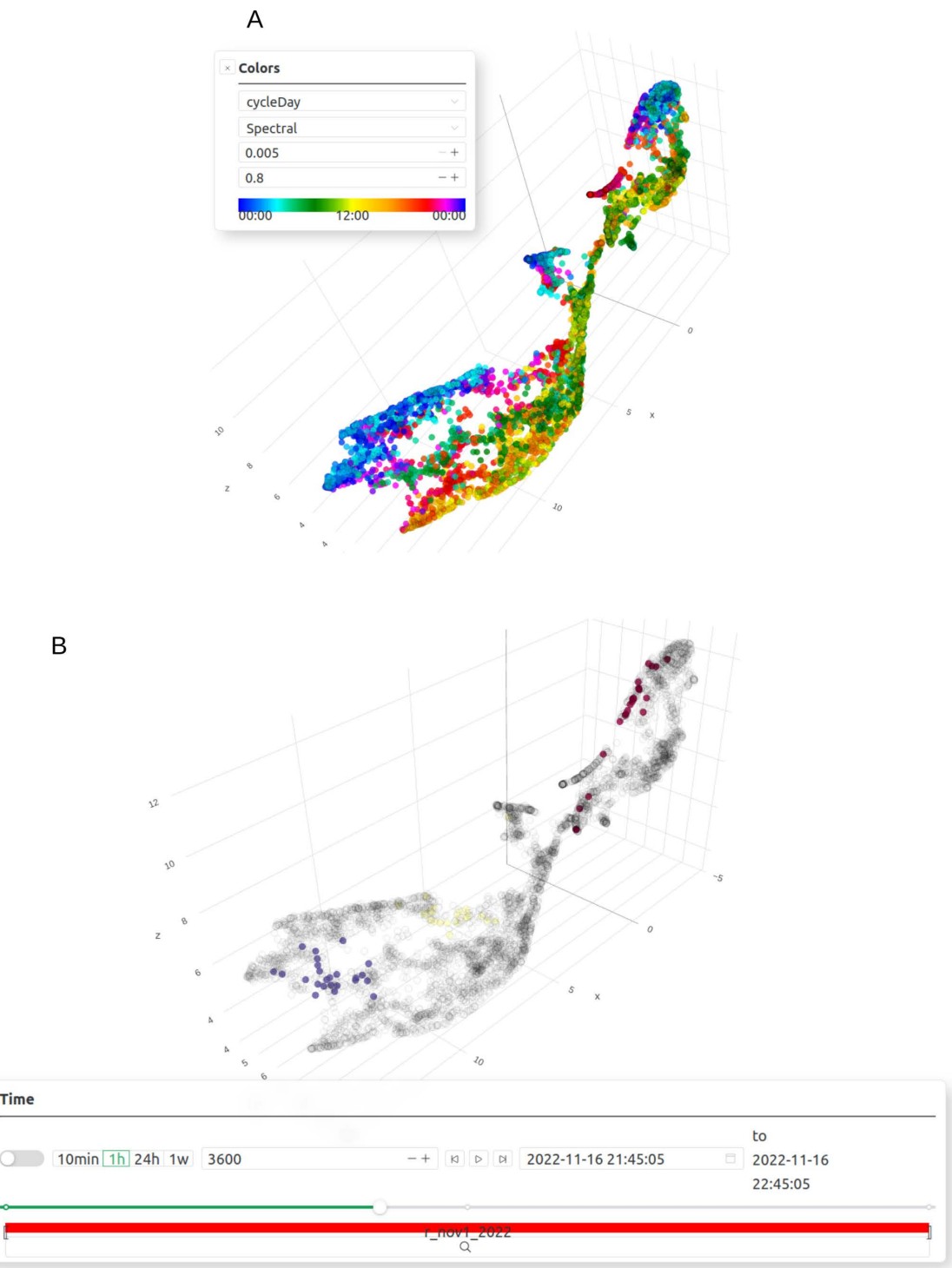

**Fig 9. Other display facilities and data exploration tools provided by** *CoralSoundExplorer* **2 (3D UMAP view). (A)** Display features: color points (sounds) using a continuous scale according to the starting time of each point (starting time of the recording). Export possibilities: UMAP/PCA plot (.png,. svg). **(B)** color only points corresponding to sounds recorded in a specified time interval. Export possibilities: UMAP/PCA plot (.png,.svg).

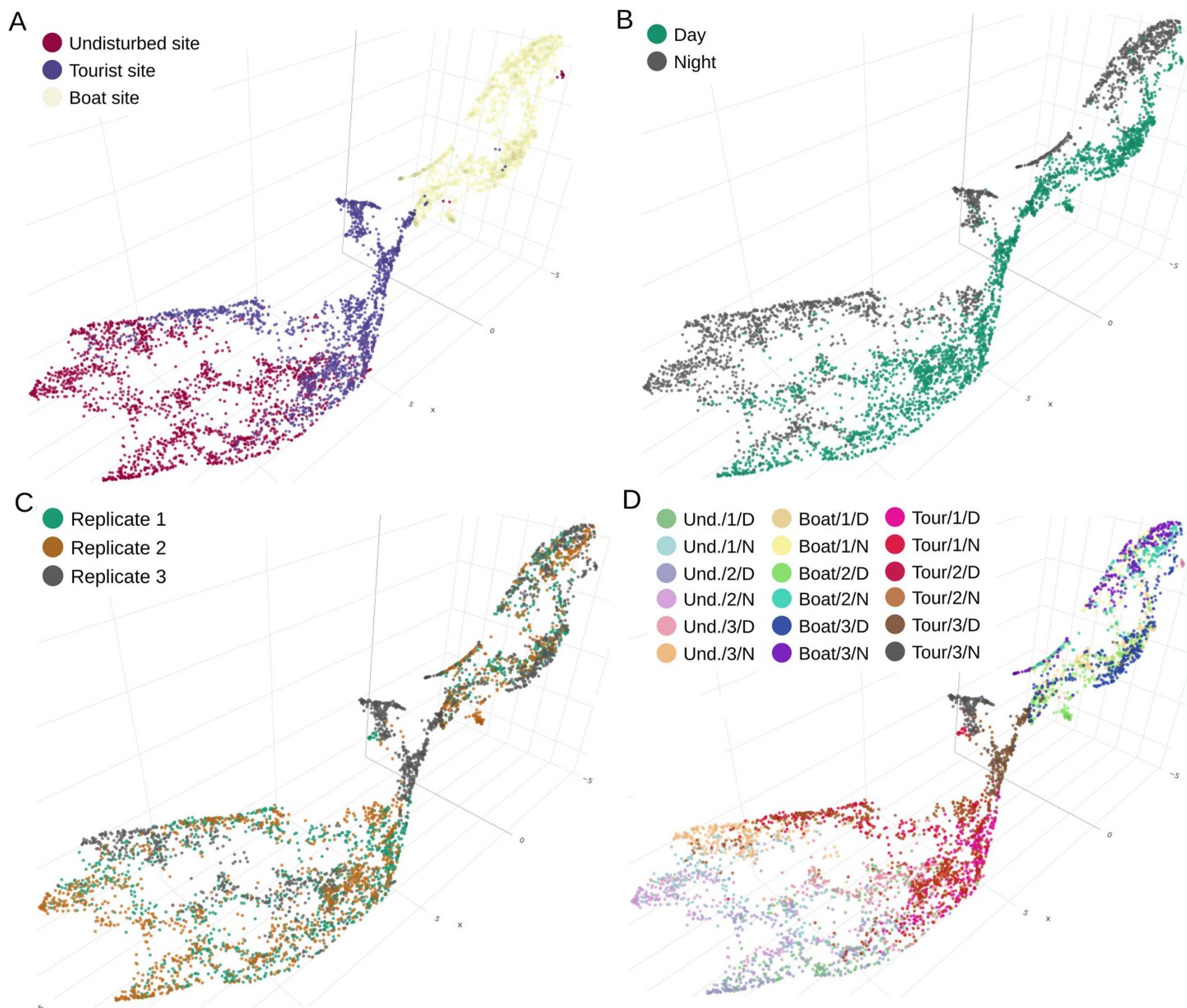

**Fig 10. Three-dimensional visualization of the acoustic space of coral reef soundscapes using *CoralSoundExplorer* software.** Each dot corresponds to 15 seconds of sound recording. Recordings were made at three sites on the Bora-Bora reef (undisturbed site, tourist site and boat site), over 24-hour periods, defining a day period and a night period, repeated over 3 non-consecutive 24-hour days (replicates 1, 2 and 3). In panels **(A)**, (B) and **(C)**, each sound is colored according to one of these predefined labels. Panel (D) combines the three labels using the following nomenclature site/replica/period (site: undisturbed as Und., tourist as Tour., and boat as Boat, replicate number 1, 2, or 3, day and night as D or N) to form 18 predefined clusters (3*2*3 = 18). These graphical representations make it easy to explore soundscapes and qualitatively grasp their correspondence with pre-defined labels. *CoralSoundExplorer*'s interface is interactive, allowing the 2D or 3D representation to be oriented as desired and zoomed in on areas of interest. Each dot is associated with the corresponding sound recording, which can be listened to and exported with a click.

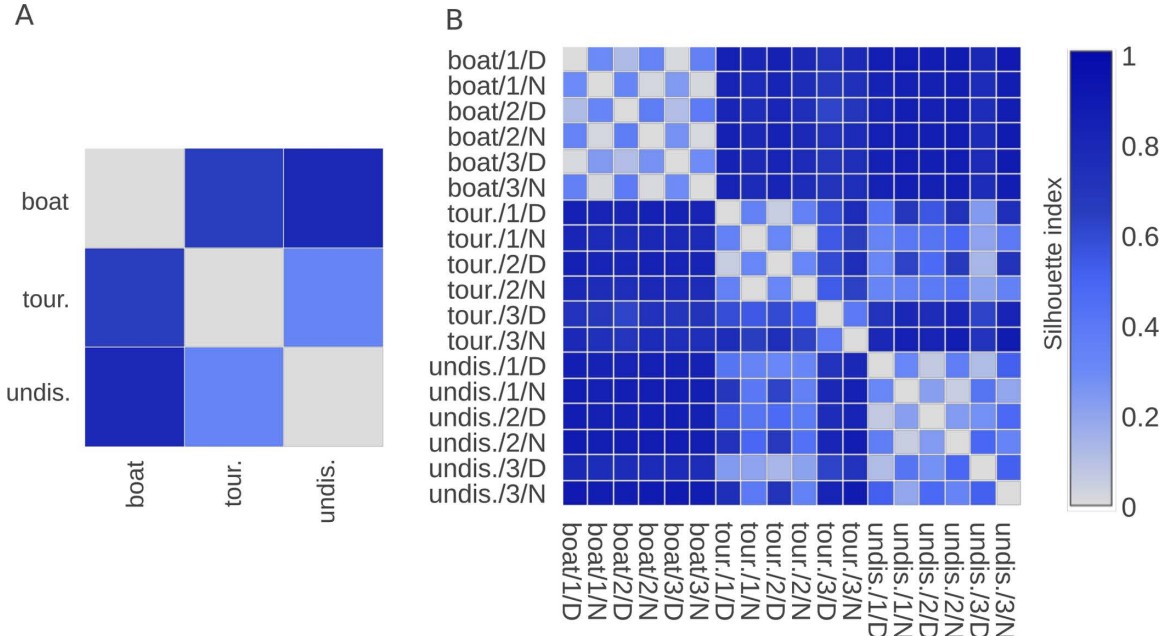

**Fig 11. Quantification of acoustic similarity (silhouette indices) between reef sounds recorded at Bora-Bora.** Silhouette indices are calculated from 100 3D UMAPs. The continuous color scale represents the index value (0: the two groups are similar, signifying homogeneous soundscapes; 1: the two groups are completely dissimilar). **(A)** Recordings are labeled by site only (tour.: tourist, boat: boat site, und.: undisturbed). **(B)** Recordings are labeled by site, day/night period (D: day, N: night) and replicate number (3 replicates, corresponding to 3 non-consecutive 24-hour recording periods).

adjusted, the overall sound level increased or decreased, as well as the playback speed. This makes it possible to inspect any sound signal in the acoustic space on demand (by listening and viewing the spectrogram), for example to understand what in the sounds' spectrotemporal structure may have influenced *CoralSoundExplorer*'s process of grouping points.

In addition to the temporal trajectory of the soundscape analysis, it is possible to obtain a quick overview of the phenology of sound samples using a continuous color scale of points according to the corresponding start time of the sample recording. In order to focus on the phenology of soundscapes defined by one or more categorical variable values, this functionality can be combined with the ability to color only the sound projection points in the projection space corresponding to one or more categorical variable values (Fig 9A).On the UMAP/PCA visual projection, *CoralSoundExplorer* also offers the option of highlighting (or color) only those points corresponding to recordings made within a specified time interval, giving the date and time of the start of this interval and its duration (Fig 9B).

## 4. *CoralSoundExplorer* software: Results obtained with the Bora-Bora dataset

The Bora-Bora dataset illustrates the main objectives of *CoralSoundExplorer* [56]: 1) visually explore a bank of sound recordings bearing labels (recording locations, date, time) and quantify the acoustic proximity of recordings as a function of these labels, 2) explore a bank of sound recordings without using labels a priori by identifying acoustic clusters, then seek to explain these acoustic clusters as a function of labels, 3) visualize and quantify the temporal dynamics of soundscapes, enabling the detection of disturbing events, and compare the temporal dynamics of different soundscapes.

For this study case, we chose to set the parameters of *CoralSoundExplorer* as follows. The initial acoustic projection space was the VGGish embedding considering the frequency band from 70 to 2000 Hz (useful frequencies of fish sounds

also covering boat noises). The final number of dimensions of this initial acoustic space retained after deleting unnecessary dimensions was 119. The integration time chosen was 15 seconds. This averaging results in 4 points per one-minute recording (instead of the initial 60 points per minute). This 15-second average was chosen because we considered it an appropriate time frame to capture both the overall acoustic environment and more punctual events such as boat noises. Because we have recorded for 24 hours from 12:05 p.m. to 12:05 p.m. (included) the next day, on three sites for three days, the entire Bora-Bora dataset was composed of 5220 samples (4*3*(6*24+1)*3). The dimension reduction process for both visualization and the computation of the distance matrix and the relative distances was the 3D UMAP. The distance matrix and the relative distances were obtained from an average of 100 UMAPs. We set the HDBSCAN algorithm to proceed with a minimum number of points for a cluster of 100 and the use of the "Leaf" and "excess of mass (EOM)" final classification algorithms [77] (see S1 Text).

## 4.1. Spatio-temporal distribution of sounds with labels

Fig 10 shows the projection in 3D acoustic space of the Bora-Bora recordings processed by *CoralSoundExplorer*. The set of points forming the cloud represents all the recordings in the Bora-Bora dataset. This figure illustrates the possibility of coloring the recordings' projections according to each of the predefined labels. In Fig 10A, the color of the dots identifies the recording sites. The three sites appear to have related soundscapes, i.e., they share certain sounds, but they are different, as the overlap between the soundscapes of the sites remains limited. The soundscape of the boat site, for example, shows no overlap with the soundscape of the undisturbed site. The soundscape of the tourist site, on the other hand, features numerous sound points in a region of the acoustic space also occupied by sound points from the undisturbed site. Some dots from the tourist site are also found in the cloud of boat sounds. The distribution of points may be partly explained by the presence of boats: there is a positive gradient from the undisturbed site to the boat site, which explains why the tourist site shares points with these two sites, having periods with and without boats.

Fig 10B shows the difference between daytime and night-time sound environments on the reef. This dichotomy between day and night can be seen at all three recording sites. However, a closer look at the distribution of labeled day and night points for each site reveals subtle variations between sites. A significant region of the acoustic space occupied by the undisturbed site sounds is thus a mixture of day and night labeled points. Sounds common to both periods suggest the activity of animal species during extended periods of dawn or dusk. The distinction between day and night sounds is clearest at the boat site. Here, the daytime soundscape is strongly affected by boat traffic, and contrasts with the nocturnal soundscape, which is certainly much quieter.

In Fig 10C, the dots representing the sounds are colored according to replicate. There is no obvious structuring in the acoustic space, particularly for the boat site and the undisturbed site. This suggests that the sound environments of these two sites remained relatively homogeneous between the recording sessions (which were not consecutive - see Methods). However, this homogeneity between replicates is not found for the tourist site, where the third replicate seems to stand out from the first two.

In Fig 10D, the dots are colored according to a composite label created from the three site/replicate/period labels. This makes it possible to assign identities to the sounds samples according to the 18 categories of this label, confirming in particular the singularity of the third replicate of the tourist site (clusters Tour./3/day mixed with Boat/all replicates/day).

## 4.2. Quantification of acoustic similarity between recordings using silhouette indices

To quantify acoustic similarities and dissimilarities between groups of sound samples defined by the categories of a categorical label (Fig 10), *CoralSoundExplorer* proposes the calculation of silhouette indices (Fig 11). For the analysis proposed here, we have chosen to work based on silhouette indices obtained between sites (Fig 11A and S1 Table) and between a composite label made from the 153 combinations of the 18 categories of the composite label (site, replicate, day/night - Fig 11B and S2 Table). Other matrices can be built, for example by analyzing silhouette indices between

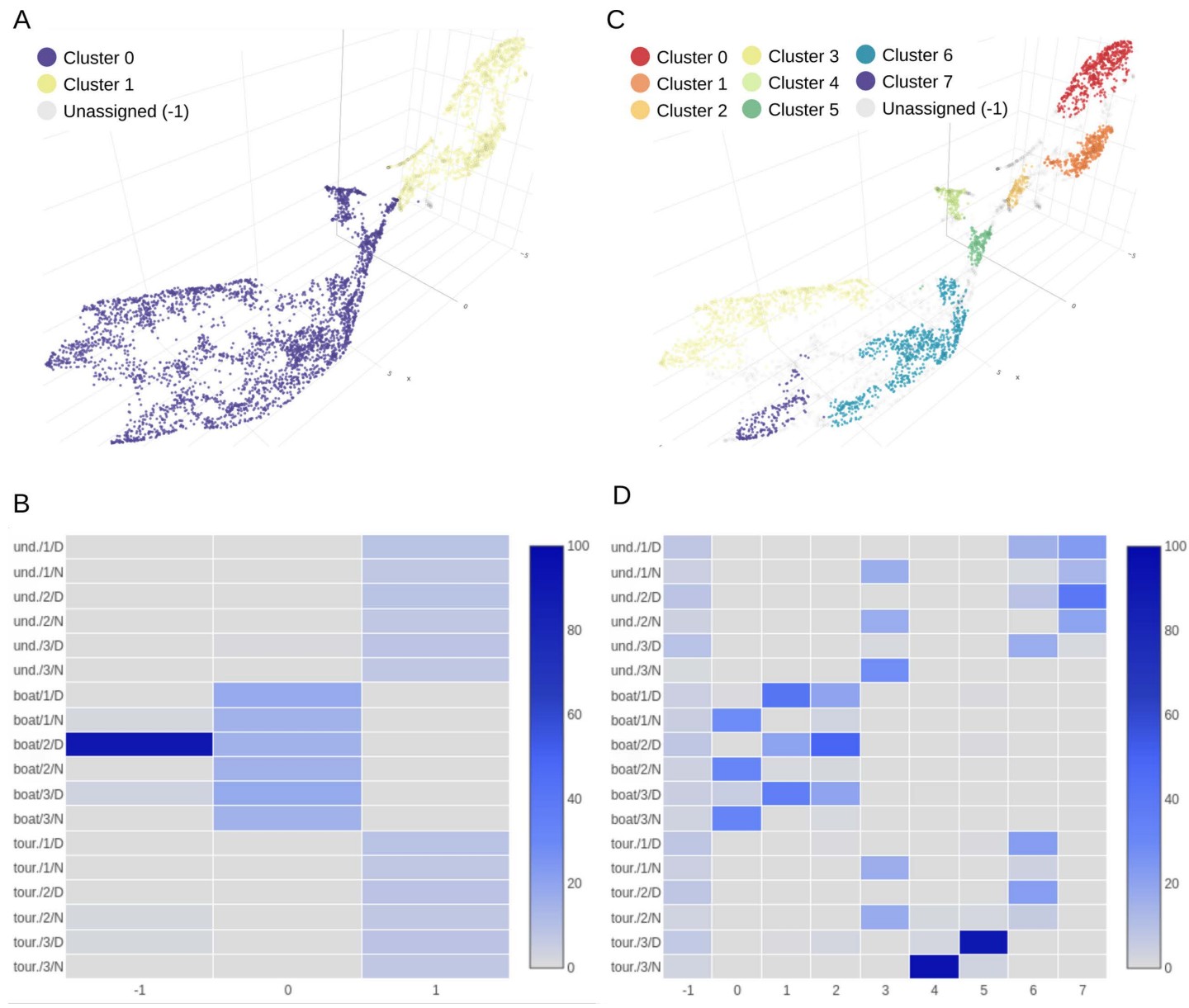

**Fig 12. Unsupervised clustering of soundscapes recorded on the coral reefs of Bora-Bora. (A)** UMAP visualization using HDBSCAN with the Excess of Mass (EOM) clustering method, which identifies two clusters separating the boat recording site from the other two sites (tourist and undisturbed). **(B)** Contingency matrix of the different categories of the composite label for each of the two unsupervised acoustic clusters obtained with the excess of mass method (EOM). **(C)** UMAP visualization using HDBSCAN with the Leaf clustering method. This method identifies eight clusters. These clusters can then be linked to specific sound sources (e.g., boat noise) by manually exploring the recordings (by clicking on the dots, inspecting the spectrograms and listening to the sounds). **(D)** Contingency matrix of the different categories of the composite label for each of the eight unsupervised acoustic clusters obtained with the Leaf clustering method.

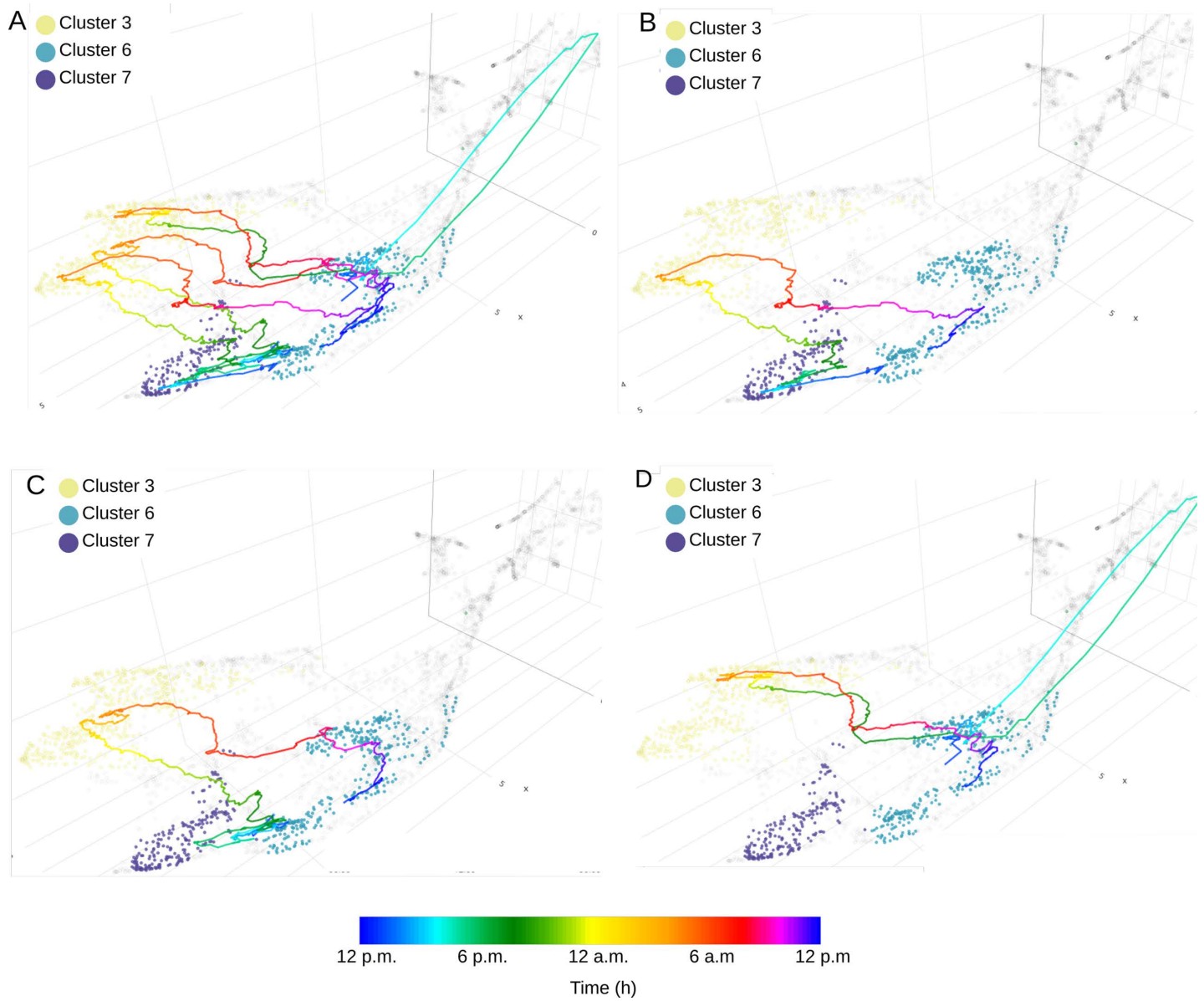

**Fig 13. Time path of soundscapes recorded at the Bora-Bora undisturbed site.** The color scale of the trajectories is related to time. (A) shows the 24-hour trajectory for three different recording days (the three replicates). **(B)**, **(C)**, and (D) show the trajectory for each replicate. Acoustic clusters were generated unsupervised using the Leaf method.

replicates or day/night periods only. A silhouette index tending towards 0 indicates a high degree of acoustic similarity between recordings, reflecting a high degree of overlap of recording clouds in acoustic space. When it tends towards 1, it means that the two categories of recordings do not share common acoustic characteristics, i.e., the clusters do not overlap in acoustic space. The data matrices shown in Fig 8 can be exported as.csv spreadsheets.

Fig 8A shows that the overall differences between sites are not identical. The silhouette indices between the boat site and the others are of the order of 0.7, while the silhouette index between the other two sites is 0.33 (silhouette indices for

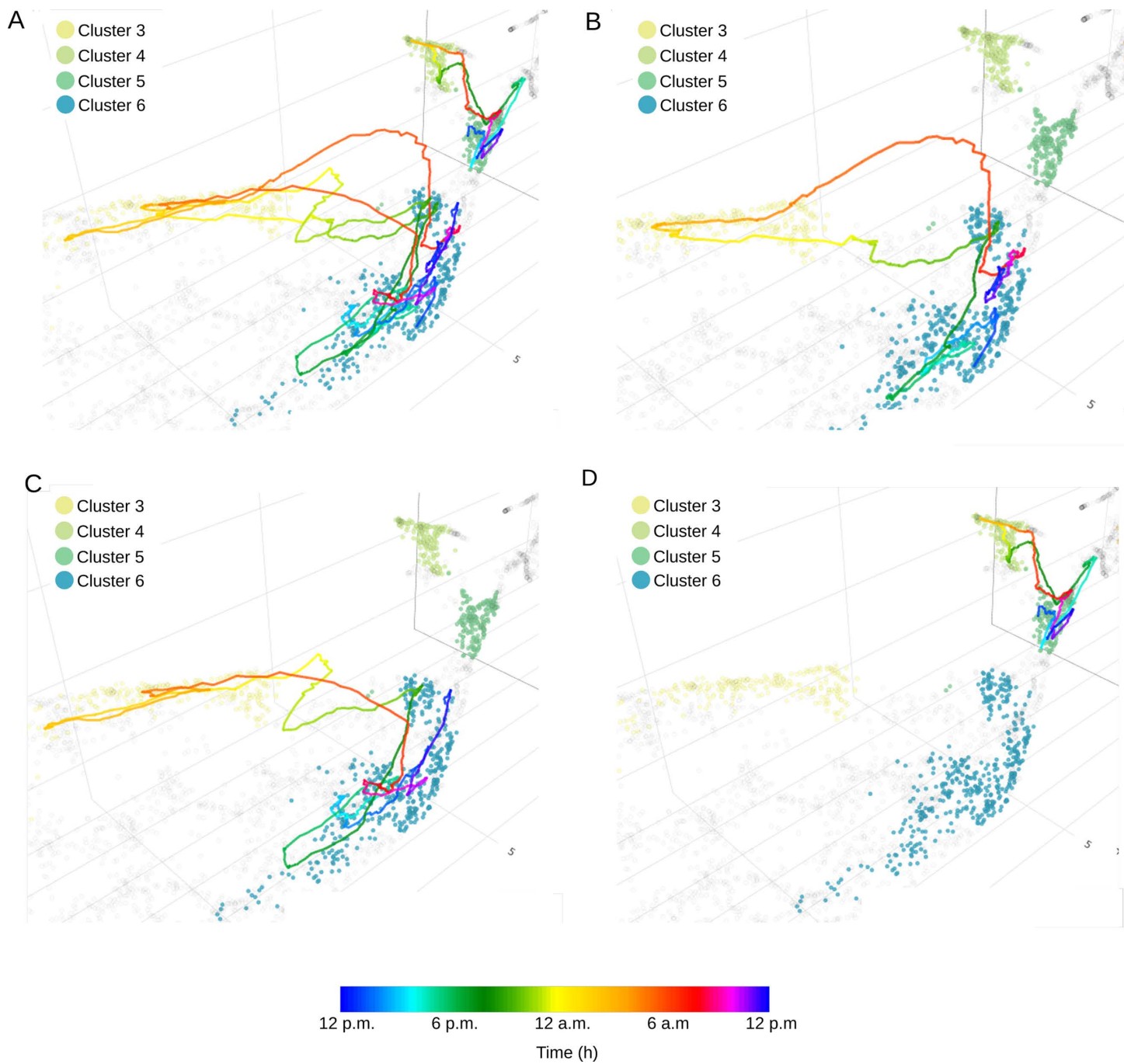

**Fig 14. Time path of soundscapes recorded at the Bora-Bora tourist site.** The color scale of the trajectories is related to time. (A) shows the 24-hour trajectory for three different recording days (the three replicates). **(B)**, **(C)**, and **(D)** show the trajectory for each replicate. Acoustic clusters were generated unsupervised using the Leaf method.

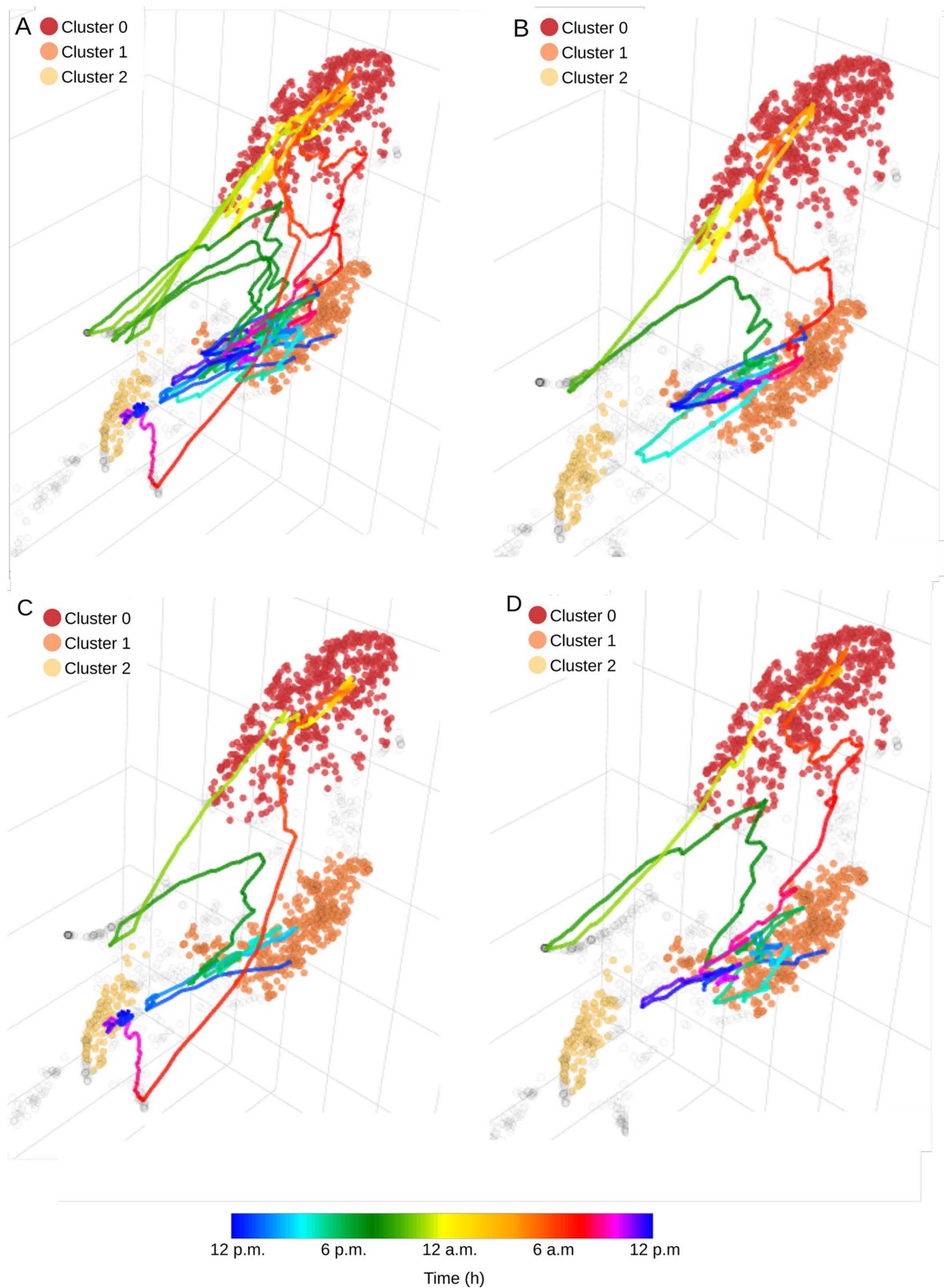

**Fig 15. Time path of soundscapes recorded at the Bora-Bora boat site.** The color scale of the trajectories is related to time. (A) shows the 24-hour trajectory for three different recording days (the three replicates). **(B)**, **(C)**, and (D) show the trajectory for each replicate. Acoustic clusters were generated unsupervised using the Leaf method.

the comparison between tourist/undisturbed sites: 0.33, between tourist/boat sites: 0.65 and between undisturbed/boat sites: 0.78). This indicates that the soundscapes of the boat site are more different with the other two sites than are the soundscapes of the tourist and undisturbed sites with each other.

By looking at a finer scale of sound groups, following the composite label (Fig 8B), we observe that the silhouette index between day and night on one site are about 0.35 whichever site is concerned (silhouette indices between day and night averaged over the three replicates: 0.35 ± 0.03 for the tourist site; 0.35 ± 0.12 for the undisturbed site; 0.32 ± 0.04 for the boat site; mean ± standard deviation). This value is about the same as the silhouette index for the comparison between tourist/undisturbed sites from Fig 11A, but is twice lower than the silhouette indices between the boat site and the two other sites. In other words, it seems that the boat site soundscapes are very different from the soundscapes of the two other sites and that these differences are of a higher degree than the differences between day and night.

In Fig 11B, we also observe low silhouette indices (close to zero) between the soundscapes of the replicates at the boat site, for both day and night. This means that the soundscapes of this site remain similar over several days for both day and night separately (silhouette index for day = 0.08 ± 0.05, for night = 0.02 ± 0.003). This result is slightly less pronounced for the undisturbed site (silhouette index between days = 0.19 ± 0.11, between nights = 0.15 ± 0.09), but the conclusions drawn for the boat site are transposable to the undisturbed site, namely the similarity between nights on the three replicates and between days on the three replicas. For the tourist site, silhouette indices are also close to 0 for replicates 1 and 2, but higher for replicate 3 (Silhouette indices between recordings made during replicate 1 and those made during replicate 2: 0.05 for days, 0.04 for nights. Silhouette indexes between recordings made during repetition 3 and the other two repetitions: 0.59 ± 0.01 for days and 0.63 ± 0.01 for nights). Thus, within a given site, except for the tourist site for the third replica, the silhouette index matrix quantifies that overall soundscapes are more similar between the three nights or between the three days, than between the nights and days of each replica. The soundscapes of replicate 3 of the tourist site are about as different from those of the other two sites as they are from those of the same site during the other two recording days.

## 4.3. Unsupervised identification of acoustic clusters

*CoralSoundExplorer* proposes the use of automatic clustering algorithms to identify groups of recordings sharing common characteristics, without any *a priori* information. This approach is useful to identify clusters of sounds that may be markers of particular sound sources (e.g., specific animal, anthropogenic noise, etc.). For each identified unsupervised cluster, its composition as regards to each predefined label can also be calculated, and eventual links between these unsupervised clusters and the period, season or time of the day can be noticed.

Applied to the Bora-Bora recordings, automatic clustering using HDBSCAN with the Excess of Mass (EOM) method identifies two clusters of recordings (Fig 12A). Cluster 0 is constituted at 98.40% by sounds from the boat site (Fig 12B and S3 Table). Cluster 1 is formed by sounds from the tourist site at 48.87% and by sounds from the undisturbed site at 49.72%, making a total of 99.59%. 16% of boat site recordings have not been assigned to a cluster (id: -1). This unsupervised clustering indicates that the boat site is acoustically very distinct from the other two sites, confirming the visualization of predefined labels and the calculation of silhouette indices.

Automatic clustering using the HDBSCAN with Leaf method identifies eight clusters (Fig 12C). Clusters 0, 1, and 2 exclusively contain recordings from the boat site. Cluster 0 primarily consists of nighttime recordings (93.68%) with constant shrimp sounds and isolated fish sounds characteristic of the boat site at night. Clusters 1 and 2 share daytime

recordings (respectively 97.86% and 90.44% – Fig 12D and S4 Table). After an audio monitoring of these sounds with the CoralSoundExplorer audio server, we observed that these two clusters differ in the nature of the soundscape. Sound signals contained in cluster 2 exhibit clear emergences of motorboat sounds with distinct isolated frequency components. Cluster 1 contains more broadband background noise. Both cluster sound samples may also be composed of various fish sounds. Sounds from both tourist and undisturbed sites are represented in cluster 3 (64.95% composed of undisturbed site recordings and 35.04% of tourist site recordings) and cluster 6 (44.58% composed of undisturbed site recordings and 55.41% of tourist site recordings), which are characterized by recordings common to coral reefs. Cluster 3 primarily consists of nocturnal recordings, very rich in impulsive sounds mostly due to shrimp activity with a low background noise level. Cluster 6 (Fig 12C) consists of daytime recordings, composed of fewer impulsive sounds but with a higher presence of low frequencies. Both cluster sound samples may be rich in various fish sounds, which is likely an indicator of the presence of healthy corals at the tourist site. Clusters 4 and 5 are exclusively composed from recordings of replicate 3 of the tourist site, respectively nighttime and daytime at 94.68% and 90.54%. Both clusters are noisy, with the distinction that cluster 4 exhibits very loud shrimp clicks that are not present in cluster 5. Cluster 7 is composed of undisturbed site recordings during the first two days. These recordings are quieter than the others, with little background noise and few clicks. Clusters 4, 5, and 7, as well as the others, contain fish sounds. An equal number of recordings from the three sites were not assigned to a specific cluster (unassigned (-1), 36.06% from the undisturbed site, 29.96% from the boat site, and 33.97% from the tourist site).

### 4.4. Temporal dynamics of Bora-Bora soundscapes and identification of environmental disturbances

As presented before, *CoralSoundExplorer* offers the opportunity to visualize and quantify how a soundscape varies over time. The temporal trajectories of soundscapes for the undisturbed, tourist and boat sites respectively, for all replicates are plotted over the eight clusters found using Leaf method in Figs 13–15. With these displays, it is easy to appreciate the variations in soundscapes over the course of a day, and to see whether these variations are homogeneous between recording days. It is also straightforward to identify events that have disrupted a daily cycle.

As shown in Fig 13, circadian changes in the soundscape of the undisturbed site are very similar across the three recording days. Clusters 6 and 7 share the day time recordings of the undisturbed site. More precisely, cluster 6 corresponds to dawn and the beginning of the day, while cluster 7 corresponds to the end of the day and dusk (Fig 13A). Replicates 1 and 2 appear to follow a similar path throughout the day (Fig 13B and 13C). On the third day of recording (replicate 3), the soundscape drifted significantly compared to the first two replicates (Fig 13D). This drift was due to a rain episode lasting about an hour (around 3 p.m.). This episode disrupted the temporal pattern of the soundscape for several hours. As a result, none of the recordings from replicate 3 are to be found in cluster 7, which corresponds to the end of the day and dusk of replicates 1 and 2. The soundscape of replicate 3 returned to a similar pattern similar to the other two replicates during the night and the following morning.

Fig 14 illustrates the temporal trajectories of the soundscape at the tourist site. The soundscape at this site is less homogeneous than at the undisturbed site (Fig 14A). Cluster 3 (representative of nocturnal recordings) and cluster 6 (daytime recordings) are shared with replicates 1 and 2 (Fig 14B and 14C). Other sounds in replicate 3 form clusters of their own (cluster 4 for night recording and cluster 5 for day recording; Fig 14D).

Fig 15 shows the temporal trajectory of the soundscape at the boat site. During the day, the overall pattern is more difficult to discern than for the other two sites, probably because boat activity is rather chaotic (variation in the number and type of boats, as well as the time of day they sail; Fig 15A). Clusters 1 and 2 (corresponding to day; Fig 15B–15D) are common to all replicates but are unrelated to time of the day. During the late-night hours (from 11 p.m. to approximately 4 a.m.), which correspond to a low boating activity, the temporal trajectories of the 3 replicates all crossed cluster 0. However, the trajectory of replicate 2 deviates from the other two between 10 a.m. and 12 a.m. (Fig 15C), probably highlighting a difference in noise due to marine traffic.

To capture the trajectory of soundscapes in the acoustic space, *CoralSoundExplorer* proposes an original representation showing variations over time in the acoustic distance between the sound and the starting point (time 12 p.m.) of the soundscape's trajectory. This representation is illustrated in Fig 16. Quantitative data (measurements of acoustic distance versus time) can be exported as.csv files, e.g., for statistical analysis.

This representation makes it possible to visualize and quantify the temporal dynamics of soundscape variations in an elegant way and using a more stable representation (i.e., based on the 100 UMAPs computations). It is easy to see in Fig 16A that the soundscape of the boat site deviates sharply from its initial position at around 7 p.m., probably due to the sharp drop in boat traffic as night falls. The distance to the initial position remains high throughout the night, suggesting a different soundscape during these hours, then decreases around 8 a.m., but only for replicates 1 and 3.

Fig 16B illustrates the displacement of the soundscape in the acoustic space of the undisturbed site. Again, the distance from the initial point increases at dusk and decreases at dawn. The first two replicates are remarkably similar. For the third replicate, there is a sudden increase in distance at around 3 p.m., indicating that the soundscape changes very rapidly at this time. This disturbance, due to a rainy episode, continues until 3.30 p.m. Only then does the soundscape approach that of the other replicates at the same moment. We can also see that the typical night-time soundscape begins

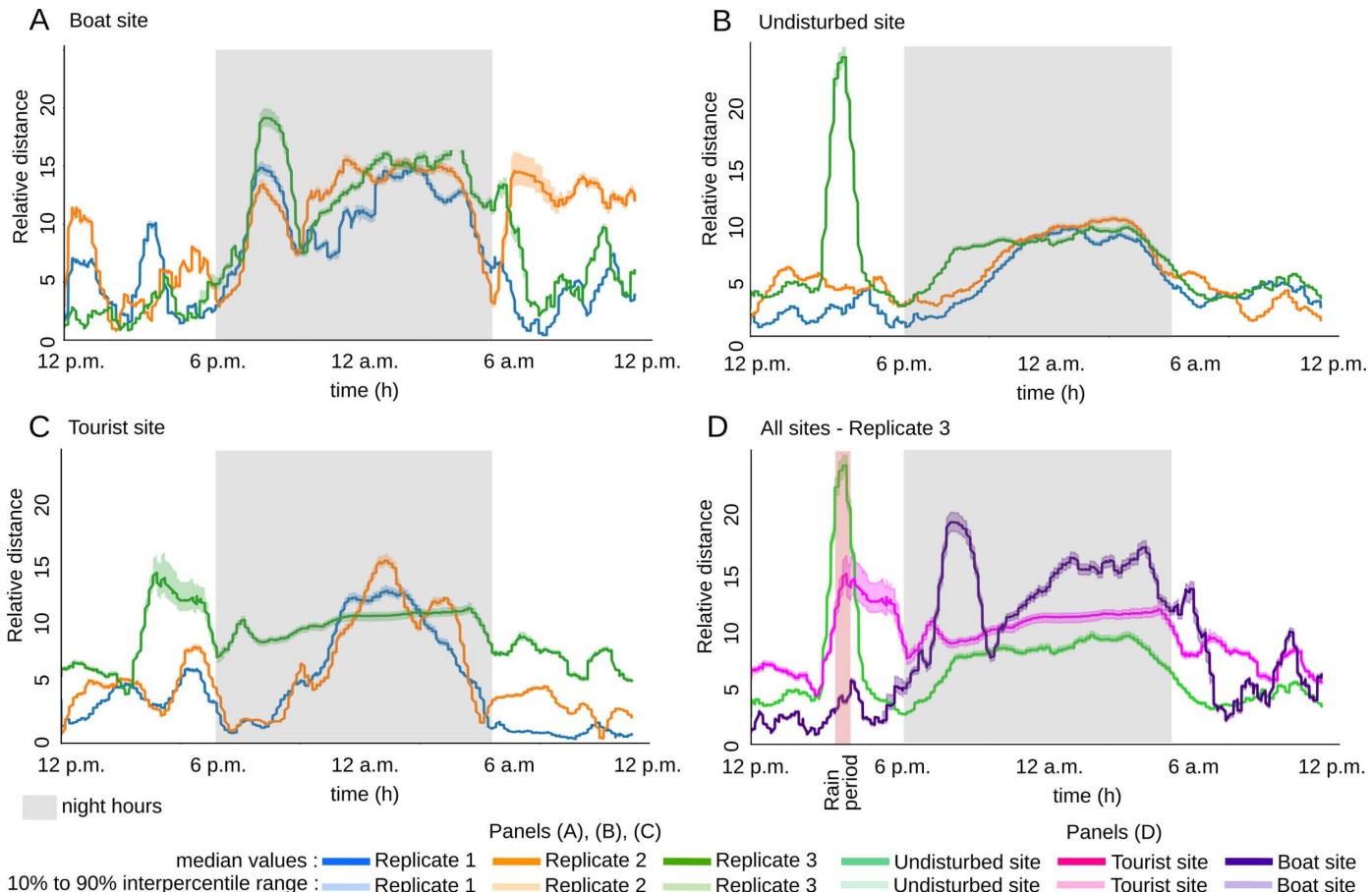

**Fig 16. Time trajectories of the three recording sites. (A)** Relative trajectories of the three replicates of the boat site over time. **(B)** Relative trajectories of the three replicates of the undisturbed site over time. **(C)** Relative trajectories of the three replicates of the tourist site over time. **(D)** Relative trajectories of replicate 3 of the three sites over time.

earlier than in the other two replicates, at around 7.30 p.m. This may be due to overcast conditions and the accompanying lower underwater luminosity.

As shown in Fig 16C, variations in distance to the initial point at the tourist site are quite similar to those described for the undisturbed site. Replicates 1 and 2 are relatively similar, and replicate 3 was disturbed at 3 p.m. due to the rain. However, unlike the undisturbed site, the tourist site did not recover its initial pattern after the rain: the soundscape is therefore further away from its initial position. It is likely that the rainy event interrupted tourist activity, and durably modified the acoustic pattern.

To facilitate comparison between sites, Fig 16D shows the variations in acoustic distances between soundscapes at the three sites on the third day of recording. Rain had an effect on tourist and undisturbed sites, and a moderate effect on boat site soundscapes, with the main peak occurring at the same time. This effect lasts a few hours for the tourist site while this is not the case for the two other sites. The undisturbed and tourist sites had a greater relative distance during this rain perturbation than the boat site, the latter being deeper, the perturbation and sound masking effect of the rain was potentially less present and also more covered by the noise of the boats.

## 4.5. Conclusion on the Bora-Bora case study

The Bora-Bora database had already been explored using conventional methods, such as manual exploration of spectrograms. This exploration, the results of which have recently been detailed [31], had required a considerable amount of time, estimated at around 15 days of work. With *CoralSoundExplorer*, we identified the same features and phenomena in just a few hours, including analysis preparation, computation time and results exploration. *CoralSoundExplorer* therefore makes it possible to rapidly track the dynamics of reef soundscapes over long periods of time.

Saving time is not the only advantage of *CoralSoundExplorer*: the visualization and quantification tools offered by our software make it possible to cluster phenomena that are difficult to detect by manual analysis. Thus, unsupervised classification identified groups of sound samples that were not distinguished by predefined labels, but differed in terms of shrimp sound quantities, fish sound quantities and types, broadband background noise levels and frequency balances, and the presence of boat noises. However, the analysis carried out here on the Bora-Bora data set did not reveal groups characterized solely by fish sounds. Numerous species of sonorous fish are probably present and active at these sites, and identifying these groups would probably enable a more in-depth ecological study. The absence of distinct groups of different fish sounds can be explained by three possibilities, not necessarily mutually exclusive. Firstly, the integration time chosen was 15 seconds, which may be a little long compared with the usual duration of fish sounds, the usual short periods of fish vocal activity and the variety of species vocalizing almost simultaneously. These isolated sounds could also be acoustically insignificant compared to the overall background noise that dominates the recordings in terms of consistency and level. A way to compensate for this problem of confounded signals in the background noise would be to use a preprocessing step of isolation of punctual acoustic events, such as the use of hydrophone arrays or numerical denoising methods. Secondly, the initial acoustic feature extractor may not be sufficiently specialized in fish sounds and *CoralSoundExplorer* could benefit from another CNN more dedicated to biological sounds. Finally, the overall acoustic differences between sound samples according to site, replicate, time of day or presence of boat and rain are greater than between the presence or absence of different fish species. Considering the entire recording campaign, the UMAPs used prior to the clustering process may have contrasted distances more according to site/replicate/period/boat/rain characteristics than according to the presence or absence of different fish species. One possibility to potentially increase the identification of fish sounds clusters would have been to work separately by site and period of day, so these categories might not have dominated the soundscape differences.

In addition to identifying site and habitat classes, UMAP plots revealed temporal patterns within data. Dynamic variations in soundscapes over a 24-hour cycle reflect both the stability of an environment (as in the case of the undisturbed site) and the arrival of particular events (rain), or variations in the soundscape between days (as in the case of the boat

site). Changes in the soundscape are thus detected, suggesting the possibility of precise monitoring of the temporal dynamics of the activity of living organisms or other sound sources. The temporal dynamics of soundscapes revealed by *CoralSoundExplorer* provides a tool for rapidly detecting and identifying disturbances and the time required to return to the initial state. For example, rainfall has a lasting effect on undisturbed and tourist sites, while the site with the highest level of human activity, and therefore noise pollution, appears to be less affected. However, for the Bora-Bora case study discussed here, we have presented a limited set of interpretations focusing mainly on the raw results provided by *CoralSoundExplorer* without going too far into ecological considerations. Further analysis could be carried out to explore in depth the relationships with biodiversity, reef health and anthropogenic disturbance. Ecological data and perhaps propagation phenomena should be integrated into the interpretation of the clusterings and the temporal patterns, and future users should ensure best practices when collecting data by controlling variables that may impact the results, such as the recorder's position, distance from the sources, or to the instrument itself.

## 5. *CoralSoundExplorer* software: A parametric study with the Bora-Bora dataset

*CoralSoundExplorer* offers various configuration options [56]. It is therefore important to analyze to what extent these parameters may influence the results in order to determine the best settings for the representation of the studied soundscapes. We firstly focused on the influence of the type of the initial acoustic space (mel-spectrum, mel-spectrogram or one of two VGGish embeddings) and the type of dimension reduction method for the visualization and the computation of the software outcomes. Considering only the use of a VGGish embedding performed on the 70–2000 Hz frequency band followed by UMAP reduction processes for both visualization and the other results computation, we also discuss the effect of the independent number of UMAPs needed for the computation part and the effect of the dimensionality of these UMAPs. It is important to know that the acoustical characteristics of a given dataset may strongly affect the results presented here. Thus, a parametric analysis should ideally be conducted systematically for each new dataset. Here we only present the conclusions deduced from these parametric studies, the related plots and the more detailed results descriptions are given in supplementary materials S2 Text.

### 5.1. Choice of the initial acoustic projection space

Regarding the choice of initial acoustic projection space, we compare the effects of the three processes available in *CoralSoundExplorer* on the Bora-Bora dataset: the mel-spectrum, the mel-spectrogram, and the VGGish embedding. For the VGGish setup, we evaluated two frequency bands: 70–2000 Hz and 125–7500 Hz. The 70–2000 Hz was also used for the mel-spectrum and the mel-spectrogram computation. This frequency band corresponds to the expected range of fish sounds and the 64 mel band filters used by these three methods fully cover this range. The 125–7500 Hz band also uses 64 mel filters, but frequencies below 125 Hz, which may be useful, are excluded. Additionally, 30 of the 64 mel bands fall outside the presumed useful frequency range. However, because the VGGish network was trained on a YouTube's video dataset over the 125–7500 Hz frequency range, it is reasonable to investigate whether the VGGish embedding performs better when input mel-spectrogram settings match those used during its training.

To compare these four acoustic projection methods, we applied the UMAP dimensionality reduction process (with 100 averaged 3D UMAPs) to the output of each and analyzed the main results provided by *CoralSoundExplorer*: the clusterability of predefined labels, unsupervised cluster identification, and temporal monitoring of the soundscape. The outcomes of the clusterability of predefined labels, the different silhouette matrices, are globally identical with only small differences that are not leading to strong divergences about the possible conclusions. The unsupervised cluster identification leads to a different number of clusters considering the different initial acoustic spaces, but these four clustering and their composition in terms of proportion of the site/replicate/period predefined labels are consistent. The temporal monitoring of the soundscape also gives mostly identical results. Considering only the undisturbed site, the soundscapes of each day follow the same relative dynamic whichever was the initial acoustic space. The two first days appear to be identical, while the

third day varies from the two others, especially for a short period of time due to a rain event. However, although this deviation is observed for the four initial acoustic spaces, its relative importance is more marked with the VGGish 70–2000 Hz embedding and the mel-spectrogram. Although different initial acoustic projection spaces produce slightly different results, the overall conclusions regarding distinctions and similarities between recording days, sites, and diurnal/nocturnal periods remain consistent. This result may seem surprising, as VGGish was designed to capture more complex sound structures than raw spectrum and spectrogram. However, the VGGish network was trained on a dataset that likely contained very few, if any, coral reef sounds. Consequently, it may rely primarily on basic spectrogram features to project these soundscape samples.

From this parametric study about the choice of the initial acoustic space, it seems that no initial acoustic space is significantly better than another. However, as the effect of rain is even more marked on the temporal trajectories using the mel-spectrogram or the 70–2000 Hz VGGish embedding, one of these approaches should be preferred. The similarity between these two initial acoustic spaces means also that 70–2000 Hz VGGish embedding compresses the mel-spectrogram representation without a significant loss of information, from 6400 to 119 features. These two comments oriented our choice to work with the VGGish 70–2000 Hz for the Bora-Bora dataset.

The results presented above not only provide guidelines for configuring the software but also offer ecoacoustic insights about coral reef sounds. These soundscapes appear to be predominantly characterized by the general acoustic background, rather than short isolated events as fish vocalizations. Thus, global spectra, especially when averaged over 15 seconds which may even more diminish the influence of isolated events, are sufficient for making distinctions or links between day/night, the recording site and the recording days and to detect soundscape deviations such as rain events.

## 5.2. Choice of the dimension reduction process

The method of dimension reduction applied to the initial acoustic projection space can also be questioned. By default, *CoralSoundExplorer* uses UMAP, a non-deterministic approach. As previously mentioned, the distance matrix between sound samples required for the HDBSCAN algorithm and the silhouette matrix is computed by averaging results from multiple UMAP iterations. Here we chose to set the number of iterations to 100. Alternatively, PCA can be employed for 3D visualization and/or for the distance matrix. Another option is to calculate the distance matrix directly from the initial acoustic projection space. Because PCA and the direct VGGish embedding process are deterministic, they eliminate the need of an averaged distance matrix, consequently one realisation of them is enough to have the distance matrix.

Comparing the 3D UMAP, 3D PCA, and direct VGGish 70–2000 Hz approaches, the silhouette values for the site/replicate/period label show for the 3D UMAP more contrasted values, more marked differences between the categories of this label. 3D PCA and direct VGGish embedding are more similar within each other. They also both reveal more similarities between a few categories of this label. The similarity between PCA and direct VGGish embedding is not surprising, as 3D PCA performs a linear transformation of the VGGish features, and the three most significant dimensions explain a not so small proportion of variance: approximately 45%. The number of unsupervised clusters identified with 3D PCA and without dimension reduction is significantly reduced, yielding only 3 and 2 clusters respectively while 3D UMAPs produced 8 clusters. However, the contingency matrices comparing these clusters to the composite site/replicate/period predefined label demonstrate logical results. The clusters observed with the 3D UMAPs process are subdivisions of the clusters obtained by the other approaches. The temporal dynamics of soundscapes at the undisturbed site over the three recorded days show qualitatively consistent results across the three approaches. The first two days exhibit similar dynamics, while the third day diverges during the rain event. However, the 3D UMAP approach highlights a greater relative amplitude of this deviation compared to the other methods. The superiority of the UMAP reduction process to distinguish between the site/replicate/period categories and to highlight an isolated soundscape deviation drove our decision to work with the UMAP reduction process. This superiority of using the UMAP on the initial acoustic space (VGGish embedding) is not so surprising. The initial acoustic space is high dimensional and the data benefit from this situation according to the concept

of "the blessing of dimensionality" which stands that in high dimensional spaces data are more linearly separable [78]. However, this effect is balanced by the "curse of dimensionality" [79] which stands that high dimensional data are sparse and that this leads to difficulties in finding potential separation between data. To deal with the "curse of dimensionality", a dimension reduction process is often used but the PCA is not based on the notions of distances and separability while the UMAP is based on them by trying to keep the distance relations and is known to increase the clustering effect [80].

## 5.3. Choice of UMAPs parameters

### 5.3.1. Number of UMAPs dimensions.

A systematic question that needs to be asked in the case of dimension reduction, using UMAPs or any other dimension reduction technique, concerns the final number of dimensions: how many dimensions are sufficient to retain most of the information? To answer this question, we have calculated two metrics of topological conservation properties as a function of UMAP dimensionality. The first, more related to the local conservation property, is the Degree of Local Preservation (DLP) [81]. The second, related to the global conservation property, is Random Triplet Accuracy (RTA) [82]. We also studied the effect of the number of UMAPs dimensions on results given by *CoralSoundExplorer*. For this study as for the DLP and RTA metrics, the following results were obtained from average distance matrices processed from 100 independent UMAPs computations. We considered four possible dimensionality values: 2D, 3D, 5D and 119D. The dimensionality of 119 is the same as that of the initial VGGish embedding, considering only those dimensions that actually support the information, so that in this case the UMAP algorithm does not strictly perform a reduction but simply transforms the space according to the UMAP principles. The dimensionalities of 3 and 2 allow us to visualize the embedding. The dimensionality of 5 is an intermediate representation.

Considering the RTA and the DLP metrics, only the 2D UMAPs is less informative, thus any of the 3 highest dimensionalities (3D, 5D or 119D) considered should be preferred to compute the 100 UMAPs. Considering either the silhouette matrices obtained, the unsupervised clustering results or the soundscapes time trajectories, no effect of the UMPAs dimensionality was observed. Thus, the 3D UMAPs reduction method was retained to work with the Bora-Bora dataset because this low dimensionality is sufficient to keep the most possible of the data information, considering the RTA and DLP, and is consistent with the 3D visual plot offered by *CoralSoundExplorer*. This result of relative independence of UMAP dimensionality on final topography is like those of a previous study in the context of using UMAPs to study communications in cetaceans [83].

These results also give a little bit more insight into the UMAP process. The fact that the 119D UMAPs works almost as well as the 3D UMAPs shows that even if the UMAP is not used for a strictly speaking dimension reduction but just as a data transformation it's almost as effective in diminishing the effect of the "curse of dimensionality" than a low dimension UMAP.

### 5.3.2. Number of independent UMAP computations.

We have determined the number of independent UMAP processes required to smooth the stochastic effect of unique UMAP using averaged distance matrices. The choice of using distance matrices to assess the stability of metrics derived from UMAP stems from the fact that the absolute values of the coordinates of UMAP make little sense for studying the relative organization of data points (i.e., sound samples). Only distances between points are necessary. We also included a variation in the dimensionality of the UMAPs considering the same four cases as in the previous section: 2D UMAPs, 3D UMAPs, 5D UMAPs and 119D UMAPs. This crossed parametric study was made because of the very small effect of the UMAPs dimensionality observed previously considering only the 100 UMAPs results. We wanted to know how the UMAPs dimensionality could influence the convergence of the successive distance matrices averaging. For each UMAP realization, the distance matrix of the data points was computed using the Euclidean distance, since the UMAP space projection is Euclidean. For the $n^{th}$ UMAP embedding, an average distance matrix was then computed from the distance matrix of the $n^{th}$ UMAP embedding, and the previous $n$-1 distance matrices. To assess the stability of the $n^{th}$ average distance matrix as a function of $n$, we calculated the relative mean absolute difference and the relative maximum absolute difference between the pairwise values of the $n^{th}$ average distance matrix and the $n$-$1^{th}$ average distance matrix.

Whatever the number of UMAP dimensions used to calculate the distance matrices, the relative mean and maximum absolute differences between the average matrix $n$ and $n$-1 decrease exponentially with $n$, demonstrating the convergence of the average distance matrix as the number of multiple UMAP transformations increases. This result is an example of the application of the central limit theorem which, assuming that the positions of data points in UMAP are bounded and therefore also the values of the distance matrix, states that this convergence should follow a $1/\sqrt{n}$ law. The convergence speeds of this metric as a function of $n$ appear to be roughly the same, with perhaps a very small advantage for the cases of higher dimensional UMAPs. The stability of the convergences of the UMAPs mean distance matrices with $n$ does not seem to be equal regarding the UMAPs dimensionality, in particular the 2d UMAPs which provided a less smooth decrease in the relative mean error curve. In terms of absolute values, the 119D UMAPs has a ten times higher absolute maximum deviation than the 3 other possibilities. The mean absolute deviation is comparable between the four UMAPs dimensionalities with comparable orders of magnitude across the four UMAPs dimensionalities either for n=1 or for n=100. This indicates that a UMAP 119D leads to less stability in the relative position of a few samples, but that these samples are so few in number that they do not strongly affect the overall organization of the whole dataset.

Besides the convergence of the distance matrices, we wanted to know if both this dimensionality and the averaging process has an impact on the relative repartition of the distances within the average distance matrices, i.e., if these parameters influence the effect of the UMAP on the reduction of the "curse of dimensionality". To address this question, we calculated a concentration metric of the pairwise distances. For each of the $n$=100 successive mean distance matrices previously obtained, we calculated the interquartile range of the pairwise distances divided by their median (IQM). The idea was to check if the UMAP process effectively increases the pairwise distance contrast and if the averaging process of distance matrices obtained from independent UMAP does not decrease this effect if it exists.

For all UMAPs dimensionalities considered, after a small increase, the IQM values are rapidly stable according to the number of successive UMAPs computed for the mean distance matrix. Compared to the distances in the VGGish embedding, the UMAPs process double the IQM even after only one realization. This proves that the UMAP process is effectively reducing the sparsity of the projected samples and that the averaging process of distance matrices does not reduce the distance contrast. The 119D and 5D UMAP transforms led to a smaller value of this contrast metric than 3D and 2D UMAP transforms, thus a very low number of dimensions for the UMAPs transform process increased the separation effect between points.

## 6. *CoralSoundExplorer*: Main achievements, limitations and perspectives

### 6.1. A powerful software tool for visualizing and quantifying coral reef soundscapes

Although there are many tools and metrics available today for analyzing soundscapes, until now there has been no software capable of visualizing them in a 2 or 3 dimensional space while associating relevant quantitative measurements. *CoralSoundExplorer* is the answer to this need [56]. Suitable for non-programming-expert users, while remaining open to modification by experts, *CoralSoundExplorer* is a tool to efficiently and rapidly observe soundscapes organization of coral reefs. A powerful visual and quantitative tool, *CoralSoundExplorer* provides a low-cost solution to coral reef monitoring and management problems.

Based on spectral or time-spectral sound characteristics, potentially through a deep learning process, *CoralSoundExplorer* avoids the often time-consuming and ambiguous process of extracting conventional acoustic indicators, as is typically the case in soundscape studies [40]. *CoralSoundExplorer* is structured into two distinct parts: the resource-intensive computational part, which concerns pre-processing, the initial acoustic embedding process, UMAP and the calculation of all results, and the visualization part, which uses a lightweight dedicated interface, directly available in a web browser but which does not require an internet connection when in use. The calculation part is fed by sound files and requires the user to fill in a spreadsheet in ODS or XLSX format. *CoralSoundExplorer* provides a template for this spreadsheet, which must list all original files, their predefined labels and computation parameters (default values are proposed). The calculation

section produces a data file (HDF5 format, [84]) containing all the results of the computational part and used to display the results. The functional separation between calculation and results visualization enables tasks to be shared between different people working on the same project.

The visualization tool uses color-scaled matrix displays, 2D or 3D map visualizations of colored points and 2D metric graphs. The sound signals corresponding to each point on the 2D or 3D visualization map can be listened to and their spectrograms visualized directly via the *CoralSoundExplorer* interface after connection to the audio module. These features make *CoralSoundExplorer* an immediately usable and easy-to-use tool for the visual analysis of sounds from a field recording campaign without extensive knowledge of machine learning. For instance, visualizing the phenology of the soundscape through paths in a 2D or 3D representation enables a monitoring operator to easily identify aberrant paths. In addition to visual representations, the results of the various analysis processes (matrix values, unsupervised cluster id, etc.) can be exported to a text file for external processing.

### 6.2. Limitations of *CoralSoundExplorer*

The results of the parametric analyses conducted here (for instance on the number of UMAP dimensions or the initial acoustic space projection method) are only really valid for the dataset used and unsupervised learning methods are sometimes difficult to handle and still require a non-naive point of view when interpreting results and adjusting parameters. Although coral reef soundscapes all share common characteristics, it is possible that for datasets larger and more complex, the values of 3 dimensions and 100 UMAPs are not suitable for a good representation of data organization and convergence of these representations. Also, the question of the UMAP, spectrogram representation or HDBSCAN hyper parameters have not been investigated in this study while they may have an influence on the results. This question of parameters or hyperparameter study is one of the main difficulties when working with unsupervised learning methods such as proposed by *CoralSoundExplorer*. First because it could be confusing when working with a "blind turn-buttons" approach especially when multiple parameters are involved and could lead to cross effects. Secondly, because the unsupervised approaches often have no exact "ground truth" to rely on, it's difficult to completely confirm the validity of the results obtained. This difficulty of dealing with unsupervised learning techniques is particularly well illustrated by the identification of unsupervised clusters. In the case of the *CoralSoundExplorer* we chose to work with HDBSCAN algorithm. As other clustering algorithms, one of the main difficulties arises from the fact that it can give different results depending on its parametrization. Although here, the two different parameters of the final clustering method ("EOM" or "Leaf") lead to results consistent with the spatio-temporal analysis of the recordings, it is possible that with a different parameterization, a finer or coarser clustering would have emerged and also made sense, or on the contrary, would have produced meaningless subgroup clusters. To evaluate the consistency of the clusters, it is always advisable to go back to listening to the sounds or visualizing their spectrogram. However, even if this listening possibility can help the operator to validate the clustering results, the acoustic characteristics used by the software to distinguish or not sound samples remain hidden, even more with the use of a CNN embedding.

The parametric study also gave a good estimation of the efficiency of each workflow step. The choice of the UMAP processes seems to be an important element for the unsupervised analysis developed for *CoralSoundExplorer*. The VGGish network can effectively give important insight about similarities and differences between coral reef soundscapes and their organizations, but it appears that the mel-spectrogram could perform almost identically as initial acoustic embedding. A network more dedicated to biological sounds detection in natural recordings, such as birdNet [85] or Perch [86], could provide a more suitable embedding for natural soundscapes study [87], more optimized for animal vocalizations, and thus give more information on the composition of soundscapes.

### 6.3. Perspectives

The choice of using 100 UMAPs, for the study case presented here, to obtain convergent quantitative results is quite strict in terms of permitted deviation from the mean and maximum errors of mean distance matrices. If the data to be processed

are large, UMAP calculations can be time and computational resources consuming. If this is penalizing, then it is possible to relax the constraints on the number of UMAPs to be averaged without leading to a significant change in the conclusions of the analysis.

UMAP is an effective method of dimensionality reduction, probably more effective than the PCA for the clustering problem. This is why we have chosen to set it by default for *CoralSoundExplorer*. However, UMAPs are not perfect in terms of topography preservation: here for example, RTA and DLP values are of the order of 40% and 50%. The expert user of *CoralSoundExplorer* will eventually be able to implement other methods based on the same principle but with a different optimization method that could be more efficient, such as triMAP or pacMAC [78].

*CoralSoundExplorer* uses the silhouette index to analyze the validity of using manual labels for data partitioning. We made this choice mainly because this measurement is widely used and relatively easy to understand. However, this metric could be discussed and there is no universal metric to assess the validity of clustering: other indices could be used. Some indices may reflect notions of clusterability different from that of the silhouette index, which is based on a notion of relative proximity. The Density-Based Clustering Validation index [88] is thus based on the notion of density and its variation between clusters. The use of a support vector machine approach [89] coupled with the adjusted mutual information index [90] could further reflect a notion of boundaries between clusters. These other indices are not implemented in *CoralSoundExplorer* but could easily be by an experienced user.

*CoralSoundExplorer*'s monitoring of sound evolution over the course of a day and in a given location is based primarily on a visual study of the paths in a 2D or 3D representation. This visualization enables an initially qualitative assessment. In quantitative terms, *CoralSoundExplorer* proposes a metric based on the extremes of the dispersions of the differences between two paths at the same time. Other metrics could be developed or derived, e.g., from the field of GPS trajectory analysis [91] or the field of time series analysis [92].

*CoralSoundExplorer* provides a novel software foundation, enabling the global exploration of coral reef soundscapes by non-programming-expert users through a user-friendly interface. As *CoralSoundExplorer* is open-source software, written in the Python and JavaScript languages, both the calculation and visualization parts can be adapted by the user to specific research needs, even outside the framework of coral reef soundscapes. For example, although *CoralSoundExplorer* currently relies on the analysis of archival data, its workflow could be adapted to real-time monitoring of soundscapes, reducing the weight of storage data and providing useful alerts for instant reef management. New analysis functions can also be added, and the CNN model used here as an option for generating the initial acoustic embedding can be replaced by another CNN that can potentially perform better than the VGGish network.

As *CoralSoundExplorer* is based on unsupervised learning, it is intended to be adaptable to different coral reefs and even other types of ecosystems. Our research team is already developing a version of *CoralSoundExplorer* fitted for other types of soundscapes or other types of bioacoustical studies (https://sound-scape-explorer.github.io/docs/). And as suggested just before, we are also working on the integration of new embeddings.

## Supporting information

**S1 Text. Detailed Methodology of *CoralSoundExplorer*: details about the software's workflow and the results computation.** Complementary material to part "II. *CoralSoundExplorer* software: Analysis workflow, graphical output and measurable metrics".
(DOCX)

**S2 Text. Parametric study of *CoralSoundExplorer*: details and results of the parametric study of the software made from the Bora-Bora dataset.** Complementary material to part "IV. *CoralSoundExplorer* software: A parametric study using the Bora-Bora dataset".
(DOCX)

**S3 Text. Software installation procedure and instructions: details about the software structure and installation instructions for various operating systems.**
(DOCX)

**S1 Table. The Silhouette indices between recordings grouped by site only: (tour.: tourist, boat: boat site, und.: undisturbed) (see** Fig 11A**).**
(CSV)

**S2 Table. The Silhouette indices between recordings grouped by the composite label site/replicate/period: site (tour.: tourist, boat: boat site, und.: undisturbed), day/night period (D: day, N: night) and replicate number (3 replicates, corresponding to 3 non-consecutive 24-hour recording periods) (see** Fig 11B**).**
(CSV)

**S3 Table. Contingency values of the different categories of the composite label for each of the two unsupervised acoustic clusters obtained with the excess of mass method (EOM) (see** Fig 12B**).**
(CSV)

**S4 Table. Contingency matrix of the different categories of the composite label for each of the eight unsupervised acoustic clusters obtained with the Leaf clustering method (see** Fig 12D**).**
(CSV)

## Author contributions

**Data curation:** Lana Minier.

**Formal analysis:** Jérémy Rouch, Rémi Emonet.

**Funding acquisition:** David Lecchini, Frédéric Sèbe, Nicolas Mathevon, Rémi Emonet.

**Investigation:** Lana Minier, Jérémy Rouch, David Lecchini, Frédéric Sèbe, Nicolas Mathevon, Rémi Emonet.

**Methodology:** Jérémy Rouch, Bamdad Sabbagh, Frédéric Sèbe, Nicolas Mathevon, Rémi Emonet.

**Project administration:** David Lecchini, Frédéric Sèbe, Nicolas Mathevon, Rémi Emonet.

**Software:** Jérémy Rouch, Bamdad Sabbagh, Frédéric Sèbe, Nicolas Mathevon, Rémi Emonet.

**Supervision:** David Lecchini, Frédéric Sèbe, Nicolas Mathevon, Rémi Emonet.

**Validation:** Lana Minier, Jérémy Rouch, Frédéric Bertucci, Eric Parmentier, David Lecchini, Frédéric Sèbe, Nicolas Mathevon, Rémi Emonet.

**Visualization:** Lana Minier, Jérémy Rouch, Bamdad Sabbagh.

**Writing – original draft:** Lana Minier, Jérémy Rouch, Frédéric Bertucci, Eric Parmentier, David Lecchini, Frédéric Sèbe, Nicolas Mathevon, Rémi Emonet.

**Writing – review & editing:** Lana Minier, Jérémy Rouch, Frédéric Bertucci, Eric Parmentier, David Lecchini, Frédéric Sèbe, Nicolas Mathevon, Rémi Emonet.

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
