## [Decision Letter · Decision Letter 0]

25 Sep 2024

Dear PhD rouch,

Thank you very much for submitting your manuscript "Visualization and quantification of coral reef soundscapes using CoralSoundExplorer software" for consideration at PLOS Computational Biology.

Your manuscript was reviewed by members of the editorial board and by an independent reviewer. In light of the reviews (below this email), we would like to invite the resubmission of a significantly-revised version that takes into account the reviewers' comments for consideration as a “Software Article”. 

We cannot make any decision about publication until we have seen the revised manuscript and your response to the reviewers' comments. Your revised manuscript is also likely to be sent to reviewers for further evaluation.

Sincerely,

Frédéric E. Theunissen

Academic Editor

PLOS Computational Biology

Denise Kühnert

Section Editor

PLOS Computational Biology

Dear Lena Minier and co-authors,

First, I want to apologize for the long time it took us to review this manuscript. I had a very difficult time finding reviewers and I casted a very wide net that included even many scientists at NOAA that are actively involved in monitoring oceans using acoustic. It is clearly a niche product and the scientist who might want to use it clearly rather be on the ocean than review papers. Since months went by and I only found one reviewer, I had to review the paper myself which is far from ideal. Your paper has many positives BUT I will classify it as a “Software Article” and not a “Research Article”.

Here are some of the strengths of the paper. First, it is clear that this will be a useful product for this community and I count on you to advertise your software and this paper. Second, I like that in the paper you stress the significance from an ecological perspective of monitoring our environments and corals in the ocean in particular. This is well described and well referenced in your introduction. Third, although your paper is long, it reads very well. The description of the methods is very clear. There are a few places where you are a bit vague (see minor comments) but you will be able to fix that. Finally, I also like you use the Bora-bora data set to illustrate the three types of analyses that are greatly facilitated by CoralSoundExplorer: supervised classification, clustering, temporal trajectories. As a Software Article, it has very few problems. I will, however, bring to your attention some issues that I find are relatively major. They have less to do with the software per se and more with thinking about how to maximize the biological impact of these measurements. I hope you can address most of these in your revision. You will see that some of the points that I bring to your attention were also raised by the reviewer (I read the paper without reading his review).

More major comments.

1. The dominant effect appears to be the recording site. Although this provides a clear, illustration that the methods “works”, it also raises the issue that the distance between sound features are dominated by “average” acoustics, which in turn are dominated by environmental factors over biological sound sources. Taking into consideration that I am somewhat of an outsider to this field, it seems that using microphone arrays and performing some sort of sound source separation before feeding the sounds to the analysis pipeline would result in a much richer analysis. This should at least be discussed and, maybe, something to add to the software?

2. To interpret the unsupervised clustering, you propose to use the audio monitoring of samples in each group. But the only interpretation your report is boat vs less boat in the boat site. This is a bit unsatisfying. I am guessing that if you had a pipeline that included sound source separation, your unsupervised clustering would be more interesting.

3. I have some worries about the use of a UMAP metric to look at temporal trajectories – as you know a non-linear embedding will distort distances – this is clearly an analysis that needs to be done with a systematic changes of some hyperparameters (see 5 below) and with a plot that provides error bars from both randomness (your 100 UMAP realizations) and sampling through hyperparameter space.

4. Your interpretation of the results for these temporal trajectories are also very descriptive with a minimum of biological interpretation. I believe this is a combination of the issues in 1 and 3. Along this point, you state in the discussion that “Dynamic variations in soundscapes over a 24-hour cycle are also highly informative.” I think that “highly informative” is a bit of an exaggeration given the observations you made here. I agree that such an analysis could be informative but here you mostly have illustrated the method.

5. Detailed Methodology should be a Method Section or a Supplementary section.

6. In the methods, I would add the equation for the mel-frequency bands that you used. Also, what is the justification for using something that was pretrained from 125 to 7500 Hz if you then apply it to a 70 to 2000 Hz. Maybe you would get better results by mapping it to fewer bands between 125 and 2000 Hz but actually matching? Or if there is some physical reason for doing that transposition because of water vs air speed of sound, you should probably justify it and use the actual physical factor.

7. Related to 6, what kind of results would you get if you fed the spectrogram (or mel-spectrogram) directly to UMAP. A great improvement obtained by using VGGish would further sell your approach.

8. I found the “conclusion” section “Main achievements and limitations” a bit redundant and the limitations that you state are mostly of computational nature (and thus quite minor). A more candid review of the more critical limitations of the entire procedure would be appropriate.

Minor Comments:

1. Author Summary l47-49. I know that you are providing a high-level overview but I think that you could be more precise in these sentences. As written, it sounds like something that one would read in an advertisement

2. Intro l100. Related to point 1, you could very briefly state what terrestrial indices are attempting to measure (and why it might not be good for aquatic environments).

3. L183. “However, such tourism access..” . You don’t need “however” here since you are not negating the previous statement.

4. L240. I would state that VGGish is trained on the sound of annotated YouTube videos.

5. As mentioned above, your treatment of the hyper-parameter fit for the UMAP is quite comprehensive. Nonetheless, it would have also been interesting to see results of actual clustering and time-courses for different dimensions. You can see that I am making this point both as more major (because of the sensitivity for the temporal trajectories) and minor here (because you did a nice job exploring the space of hyperparameters for UMAP)

6. In the legend of Fig 6, I would specify that the unsupervised cluster with label -1 is what HDBSCAN has classified as “noise”, or more precisely as having low probability of belonging to any of the clusters.

Frederic Theunissen

Reviewer's Responses to Questions

**Comments to the Authors:**

Reviewer #1: SUMMARY

The paper describes a novel software tool that uses a pre-trained VGGish network model as a feature extractor and UMAP to generate low-dimensional visualisations. UMAP is also used to compute a distance matrix for subsequent analyses - this is achieved by averaging over multiple UMAP runs. The results presented for the Bora-Bora dataset (e.g. meaningful, visually discernible clusters, state space trajectories, and contingency tables presented on an interactive interface) demonstrate the usefulness of the tool for soundscape analysis.

In fact, the relevance of the tool is broader than it might first appear: apart from a few (editable) parameter choices, there doesn't seem to be anything that would make it exclusive to coral reef environments. The generalisability to other datasets (of either same or different biomes), however, has not been demonstrated. As an example, the choice of the number of UMAP replicates was based on convergence analysis (Figure 17), but the results may be dataset-specific (as acknowledged by the authors on page 46). Applying the method to different datasets would certainly be helpful - though not a requirement. As an alternative, a better understanding of the method, in particular the distance matrix averaging procedure, could help ensure robust performance across diverse datasets.

On this topic, a major point is raised below. In addition, a number of minor comments are also offered.

MAJOR

Stochastic neighbourhood embedding methods like UMAP create low-dimensional representations that aim to preserve the distance structure present in the high-dimensional data, but this inevitably introduces distortions due to approximations (e.g. limited neighbourhood size) and stochasticity. The latter, in particular, affects reproducibility. To address this issue, the authors propose running multiple replicates of UMAP and using the average distance matrix for downstream tasks. However, this approach raises questions.

1) As is generally the case with stochastic estimation, averaging over multiple runs should yield a better approximation of the distance matrix. However, the distance matrix in the original VGGish embedding space is accessible. In fact, an estimate of it (based on neighbourhoods of limited size) is used as input to compute the UMAP embeddings. Why going through the trouble of averaging over 100 replicates of UMAP estimates of the distance matrix when the original one is accessible?

The answer to this question seems to have to do with the curse of dimensionality (and the related distance concentration problem): as the number of dimensions increases, the volume of the space increases exponentially. This leads to data points being sparse and spread out, which in turn makes distances between points less meaningful (see "On the Surprising Behavior of Distance Metrics in High Dimensional Space" by Aggarwal et al.). That poses problems for clustering, as confirmed by the authors with the Hopkins statistics for 3D (single run UMAP) vs 119D (original VGGish embedding) spaces [line 807]. Reducing dimensionality is thus a good idea. In fact, it is common practice (eg, recommended in UMAP python package documentation, https://umap-learn.readthedocs.io/en/latest/clustering.html). That brings us to the second question:

2) Doesn't the proposed procedure (ie, averaging the distance matrix over multiple UMAP runs) reintroduce the problems faced in high dimensional spaces (ie, render pairwise distances less informative)? Or is it really possible to reap the benefits of a more faithful approximation of the original distance structure while still avoiding the curse of dimensionality?

To properly address this point, a number of methods for assessing distance concentration exist. One possible analysis in the spirit of figure 17 is to measure the coefficient of variation of distances (similarly, IQR over median, and contrast of distances) of D(n), n in N = (1, 100). The plotted values are expected to be stable (good for the proposed method) or decay (not so good). The baseline for comparison should be the same metrics (CV, IQR/median, and contrast of distances) applied to the original VGGish space.

MINOR

127: The interface is described as "easy-to-grasp," but its usability has not been explicitly evaluated (e.g., with a user study). Instead of using this term, it could be stated that the interface was designed with the objective of being intuitive and user-friendly. This clarifies that ease of use is a design goal, although it hasn't been formally evaluated.

128: "AI deep integration" is not a technical term I'm familiar with. Perhaps what is meant is that the tool integrates a (large, pre-trained) deep learning model into an interactive system.

130: Typo: convolutional instead of convoluted. It might be relevant to mention that the CNN has been pre-trained on a large audio dataset.

132: Suggested rephrasing: instead of "efficiently capture...data": something along the lines of "replicate in lower dimensions the same geometric relations present in the original, high dimensional space"

150: It is not clear what is meant by "measure the metrics"

214: The frequency band of interest and sample length don't seem to match that expected by VGGish (125-7500 Hz, 0.96 s, https://github.com/tensorflow/models/blob/master/research/audioset/vggish/README.md). Can the authors please clarify this mismatch?

241: Mention what VGGish is pre-trained on - a large audio dataset.

266: Couldn't the reproducibility issue be solved simply by fixing the random seed?

275: "UMAP projections": The term "projection" can have a specific mathematical meaning that differs from how UMAP works. UMAP is based on manifold learning, which is slightly different from linear projections like PCA. While "projections" in the context of UMAP is accepted and used, alternative terms like "embeddings" might be more appropriate.

529: "Clusters ... are common at all replicates but are few related to specific daytime hours" is unclear. Is it meant that clusters are weakly/un-related to time of the day?

599: "semi/unsupervised learning methods": What is meant by semi-supervised learning in the proposed method? (See comment about line 618)

609: I am not familiar with the use of the term "individuals" to refer to data points. Data points, or samples, would sound more natural to me.

617: Typo: embedding space instead of embedded space

618: "Semi-supervised" typically refers to a form of learning that leverages both labeled and unlabeled data to build better models, especially when labeled data is limited. In this case, however, the term doesn't seem to apply, as there is no learning (i.e., no guided update of model parameters) informed by the labels taking place.

657: For a given fixed duration, the number of samples per STFT step depends on the sampling rate of the input file. Does the tool resample files to a fixed sampling rate? (See also comment about line 214)

750: "'By chance' value" is more commonly referred to as "chance level", without quotes.

752: Typo: Monte Carlo, not Monte Carlos

796: Sentence ending on line 797 is very unclear. What values could occur?

797: "Modality" often means something else. "Class," "label class," or "category" are more common terms for referring to different groups or labels in data classification.

818: "expected" instead of "supposed"

Lastly:

Fig 3: There is a mismatch between caption and figure

While VGGish is an important, general-purpose audio feature extractor, its representation space is optimised for a different set of sounds: VGGish was pre-trained on a large YouTube audio dataset, a data domain different from bioacoustics. In fact, for several biomes and groups of species, BirdNet (https://doi.org/10.1016/j.ecoinf.2021.101236) has been shown to be a better embedding space (see https://ai-2-ase.github.io/papers/35%5cCameraReady.pdf and https://doi.org/10.1038/s41598-023-49989-z). It would be informative for the readers to state the limitations of applying a model trained on YouTube, and what improvements could be expected in the future by the incorporation of a domain-specific large pre-trained embedding model.

**Have the authors made all data and (if applicable) computational code underlying the findings in their manuscript fully available?**

Reviewer #1: Yes

PLOS authors have the option to publish the peer review history of their article (what does this mean? ). If published, this will include your full peer review and any attached files.

**Do you want your identity to be public for this peer review?** For information about this choice, including consent withdrawal, please see our Privacy Policy .

Reviewer #1: **Yes: ** Thiago S. Gouvêa
---

## [Decision Letter · Decision Letter 1]

28 Feb 2025

Dear PhD rouch,

We are pleased to inform you that your manuscript 'Visualization and quantification of coral reef soundscapes using CoralSoundExplorer software' has been provisionally accepted for publication in PLOS Computational Biology.

Best regards,

Frédéric E. Theunissen

Academic Editor

PLOS Computational Biology

Denise Kühnert

Section Editor

PLOS Computational Biology

Dear Author,

Thank you for your diligence and patience. It is a very nice contribution to the field and I hope that your methods and software will be used by many bioacoustibcians.

Best wishes,

Frederic Theunissen

Reviewer's Responses to Questions

**Comments to the Authors:**

Reviewer #1: The authors have satisfactorily addressed all points raised in the initial review.

**Have the authors made all data and (if applicable) computational code underlying the findings in their manuscript fully available?**

Reviewer #1: Yes

PLOS authors have the option to publish the peer review history of their article (what does this mean? ). If published, this will include your full peer review and any attached files.

**Do you want your identity to be public for this peer review?** For information about this choice, including consent withdrawal, please see our Privacy Policy .

Reviewer #1: **Yes: ** Thiago S. Gouvêa

---

## [Editor Report · Acceptance letter]

PCOMPBIOL-D-24-00575R1

Visualization and quantification of coral reef soundscapes using CoralSoundExplorer software

Dear Dr rouch,

I am pleased to inform you that your manuscript has been formally accepted for publication in PLOS Computational Biology. Your manuscript is now with our production department and you will be notified of the publication date in due course.

With kind regards,

Anita Estes
